# ZNF524 directly interacts with telomeric DNA and supports telomere integrity

Hanna Braun [1,2,8], Ziyan Xu[3,8], Fiona Chang[1], Nikenza Viceconte[2], Grishma Rane[1], Michal Levin[2], Liudmyla Lototska[2], Franziska Roth[2], Alexia Hillairet[1], Albert Fradera-Sola [2], Vartika Khanchandani [1], Zi Wayne Sin[1], Wai Khang Yong [1,4], Oliver Dreesen [5], Yang Yang[3], Yunyu Shi[3], Fudong Li [3] ✉, Falk Butter [2,6] ✉ & Dennis Kappei [1,4,7] ✉

Telomeres are nucleoprotein structures at the ends of linear chromosomes. In humans, they consist of TTAGGG repeats, which are bound by dedicated proteins such as the shelterin complex. This complex blocks unwanted DNA damage repair at telomeres, e.g. by suppressing nonhomologous end joining (NHEJ) through its subunit TRF2. Here, we describe ZNF524, a zinc finger protein that directly binds telomeric repeats with nanomolar affinity, and reveal base-specific sequence recognition by cocrystallization with telomeric DNA. ZNF524 localizes to telomeres and specifically maintains the presence of the TRF2/RAP1 subcomplex at telomeres without affecting other shelterin members. Loss of ZNF524 concomitantly results in an increase in DNA damage signaling and recombination events. Overall, ZNF524 is a direct telomere-binding protein involved in the maintenance of telomere integrity.

Telomeres are nucleoprotein structures at the ends of linear chromosomes and are essential for genome stability. In particular, telomeres are constitutively bound by the shelterin complex, which hides telomeres from recognition by the DNA damage repair (DDR) machinery. Telomeric deprotection through telomere shortening or displacement of shelterin proteins may lead to chromosomal fusions. Such events trigger an escalating cascade of telomere-induced genomic instability via breakage-fusion-bridge cycles and chromothripsis that is observed in a large proportion of cancers[1].

Within the six-protein shelterin complex, telomeric repeat binding factors 1 and 2 (TRF1 and TRF2) form homodimers to bind the double-stranded TTAGGG repeats directly via homeobox domains, while RAP1 associates with telomeres based on its interaction with TRF2[2–6]. POT1 is linked to the complex through TPP1 and TIN2 and binds telomeric single-stranded DNA directly[3]. In particular, the shelterin member TRF2 plays a central role in telomere protection: TRF2 is essential for both the formation and preservation of the t-loop, a structure that protects the open chromosome ends by strand invasion of the 3'-overhang into the double-stranded telomeric region[3,7,8]. Crucially, TRF2 prevents ATM-mediated DDR and NHEJ at telomeres, with RAP1 supporting this function and providing a backup mechanism in the case of telomere shortening and impaired topology[3,9–14]. In addition, RAP1 has implications in the prevention of homology-directed repair (HDR) at telomeres and reduces recombination events[3,15]. While the TRF2/RAP1 subcomplex has been previously shown to be able to associate with telomeric DNA independently of the fully assembled shelterin complex in vitro, in vivo evidence for an independent regulation has thus far been lacking.

Beyond the shelterin complex, additional proteins have been described at mammalian telomeres[16–18]. Among them, a few bind

---

[1]Cancer Science Institute of Singapore, National University of Singapore, Singapore 117599, Singapore. [2]Institute of Molecular Biology (IMB), Mainz 55128, Germany. [3]MOE Key Laboratory for Cellular Dynamics, School of Life Sciences, Division of Life Sciences and Medicine, University of Science and Technology of China, Hefei, China. [4]Department of Biochemistry, Yong Loo Lin School of Medicine, National University of Singapore, Singapore 117596, Singapore. [5]Cell Aging Laboratory, A*STAR Skin Research Labs, Singapore 138648, Singapore. [6]Institute of Molecular Virology and Cell Biology (IMVZ), Friedrich Loeffler Institute, Greifswald 17493, Germany. [7]NUS Center for Cancer Research, Yong Loo Lin School of Medicine, National University of Singapore, Singapore 117599, Singapore. [8]These authors contributed equally: Hanna Braun, Ziyan Xu. ✉e-mail: lifudong@ustc.edu.cn; f.butter@imb.de; falk.butter@fli.de; dennis.kappei@nus.edu.sg

directly to telomeric DNA, such as the telomere length regulators HMBOX1 (HOT1)[19] and ZBTB48 (TZAP)[20,21], as well as alternative lengthening of telomeres (ALT)-specific factors, such as NR2C/F nuclear receptors[17,22,23] and ZBTB10[24]. Our previous phylointer-actomics screen[16] recapitulated all these telomeric proteins and suggested further candidates, including ZNF524, a 28.7 kDa zinc finger protein of unknown function (Fig. 1a), which we here demonstrate to directly bind to telomeric DNA and to contribute to telomeric stability as a positive regulator of the TRF2/RAP1 subcomplex.

## Results

### ZNF524 binds telomeric repeats directly with nanomolar affinity

To verify the association of human ZNF524 with telomeric DNA, we performed a DNA pull-down assay. Endogenous ZNF524 was enriched on TTAGGG repeats and on the telomeric variant repeats TCAGGG, TGAGGG and TTGGGG but not on the scrambled control sequence (Fig. 1b). To test whether ZNF524 binds telomeric DNA via its four Cys2His2-type (C2H2) zinc fingers, we used FLAG-ZNF524 WT and individual ZF point mutants in the DNA pull-down assay. Indeed, the ZF2 mutant failed to enrich on telomeric DNA (Fig. 1c, Supplementary Fig. 1a). To further test for the direct interaction with telomeric DNA, we expressed recombinant His-ZNF524 in *E. coli* and again observed that His-ZNF524 WT was enriched on TTAGGG and variant repeat sequences, while the His-ZNF524 ZF2 mutant only showed a residual background signal (Supplementary Fig. 1b). We next quantified binding affinities for ZNF524 with double-stranded telomeric DNA by isothermal titration calorimetry (ITC). Using a minimal domain containing only the four zinc fingers (110-223 aa), we detected binding to a 12 bp

telomeric sequence with a $K_D$ value of 90 nM (Fig. 1d, Supplementary Table 1). Similar to endogenous ZNF524, we observed binding of the minimal domain to the telomeric variant repeats TCAGGG, TGAGGG and TTGGGG, albeit with reduced affinities of 280 nM, 270 nM and 330 nM, respectively (Supplementary Fig. 1c, Supplementary Table 2). Again, an interaction with the scrambled control was not observed. As telomeres are prone to forming G-quadruplexes (G4s) and comprise a G-rich ssDNA overhang, we also tested binding of the ZNF524 minimal domain to these structures but did not detect interactions (Supplementary Fig. 1d, e). Furthermore, removing either ZF1 or ZF4 from the ZNF524 minimal domain led to strongly reduced affinities for dsTTAGGG with $K_D$ values of ~620 nM and 590 nM, respectively. Deletion of either ZF1 + 2 or ZF3 + 4 led to a complete loss of binding ability (Fig. 1d, Supplementary Table 1), indicating that ZNF524 directly binds to TTAGGG repeats, likely involving all four zinc fingers.

### ZNF524 employs four zinc fingers for base-specific recognition

To understand the interaction of the four zinc fingers with the telomeric motif in atomic detail, we crystallized the ZNF524 DNA-binding domain (107-237 aa) with telomeric DNA and solved its structure to 2.40 Å (Supplementary Table 3). In the crystal, each crystallographic asymmetric unit contained one ZNF524 minimal domain bound to a telomeric duplex DNA (Fig. 2a). In the final structural model, all the DNA nucleotides and most of the amino acids (113-225 aa) could be clearly built. The DNA molecules in the crystal are coaxially stacked, with the terminal A and T bases of neighboring DNA molecules pairing to form a pseudo continuous TTAGGG duplex. Each ZF adopts a canonical C2H2 zinc finger fold (Fig. 2a) with its recognition α-helices

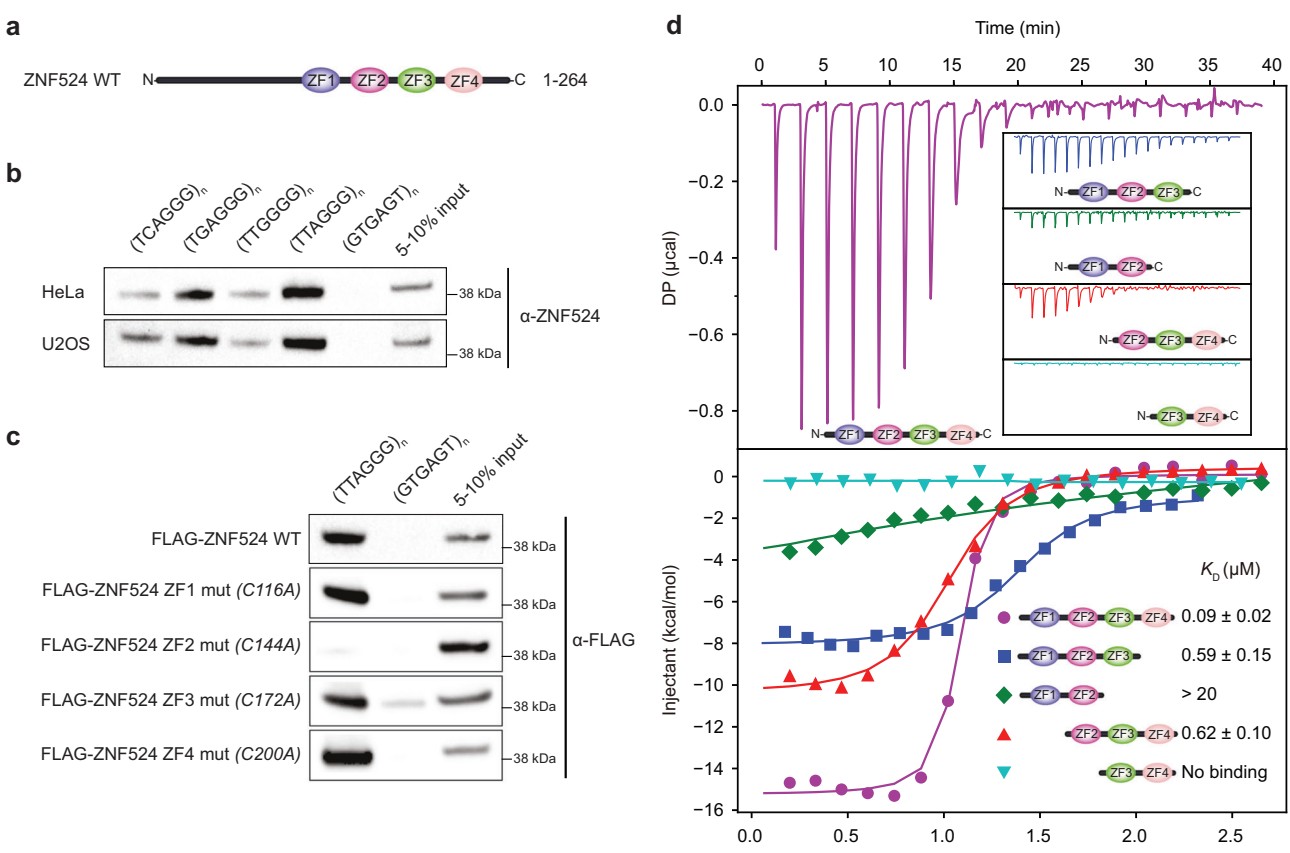

**Fig. 1 | ZNF524 directly binds TTAGGG repeats through its zinc fingers.**
**a** Schematic overview of the ZNF524 protein with its four C-terminal zinc fingers. **b** DNA pulldowns with canonical and variant telomeric repeats using HeLa and U2OS lysates (n = 1). **c** DNA pulldowns with individual FLAG-ZNF524 ZF mutants overexpressed in HEK293 cells (n = 1). **d** Isothermal titration calorimetry results for different combinations of ZNF524 ZFs with a 12-bp ds(TTAGGG)2. $K_D$ values with standard deviations are noted in the lower right corner.

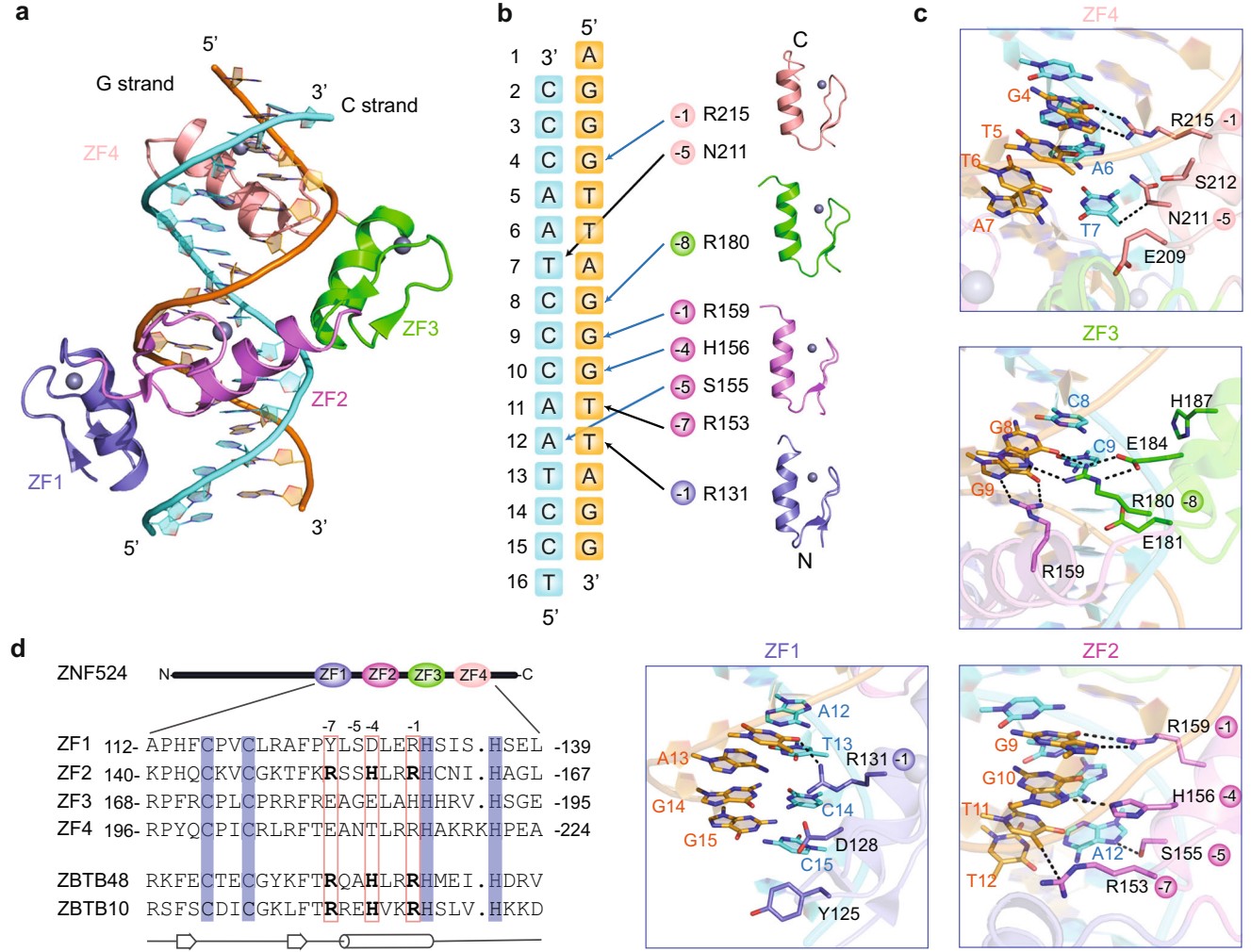

**Fig. 2 | Crystal structure reveals base-specific interactions of all four ZFs.**
**a** Overall structure of the four zinc fingers (ZF1 (blue), ZF2 (violet), ZF3 (green) and ZF4 (salmon)) in complex with duplex telomeric DNA (G-strand (orange), C-strand (cyan)), (**b**) Schematic representation of base-specific contacts of the ZFs with telomeric sequences. Hydrogen bonds (blue) and van der Waals contacts (black) are highlighted. **c** Details of the base-specific contacts for each individual ZF. The hydrogen bonds and van der Waals contacts are depicted as black dashed lines. **d** Sequence alignment of ZF1, ZF2, ZF3 and ZF4 of ZNF524, ZF11 of ZBTB48 and ZF2 of ZBTB10. The four zinc-coordinating residues of each finger are highlighted with a blue background. The first zinc-coordinating histidine in each finger serves as reference position 0 for the RxxHxxR motif (bold).

inserted into the major groove, making base-specific contacts through both hydrogen bonds and van der Waals interactions (Fig. 2a, b). In agreement with the DNA pull-down and ITC results, all four ZFs form direct, base-specific interactions with telomeric DNA. In particular, ZF2 is central to the DNA interaction with the four amino acids Arg153, Ser155, His156 and Arg159, making base-specific contacts along a 4-bp region (Fig. 2b, c). Such an RxxHxxR motif was previously also found in ZF11 of ZBTB48, the sole zinc finger required for its telomeric binding[20,25], and more recently also in ZF2 of ZBTB10[26] (Fig. 2d). Similarly, in ZNF524 ZF2, Arg159 donates two hydrogen bonds to the O6 and N7 atoms of G9, forming a bidentate H-bond interaction (Fig. 2b, c). Additionally, the Nε2 group of His156 donates one H-bond to the N7 atom of G10 (Fig. 2b, c). Furthermore, Ser155 forms hydrogen-bonding interactions with the N7 atom of A12 at the C strand. In combination with ZF1, which forms one hydrogen bond between T12 at the C strand and Arg131, this provides specific recognition to the T:A at position 12 (Fig. 2b, c). However, Arg153 forms a van der Waals contact with the methyl group of T11, deviating from the ZBTB48 structure, where it recognizes the GGG triplet. Here, the RxxHxxR motif of ZF2 recognizes a TGG triplet instead of GGG, while the GGG triplet is jointly recognized by ZF2, ZF3 and ZF4

(Fig. 2b). The combination of the base-specific contacts of all four ZFs mediates binding to the telomeric TTAGGG repeat. Mutation of any of the determined binding residues led to reduced affinity with the most drastic effect for combined mutations in ZF2 (Supplementary Table 1). In conclusion, these data suggest that the RxxHxxR motif is a common interaction feature of telomeric zinc finger proteins.

## ZNF524 localizes to telomeres in vivo
Next, we examined whether ZNF524 localizes to telomeres in vivo. Using doxycycline-inducible ZNF524-GFP in U2OS cells, we observed the formation of foci that colocalized with TRF2, which was used here as a bona fide telomeric marker. Overall, in >90% of cells, colocalization was visible, with an average of 66% of all TRF2 foci being positive for ZNF524 (Fig. 3a, b). In agreement with the in vitro data, the DNA-binding deficient ZNF524-GFP ZF2 mutant (C144A) did not form foci and was distributed diffusely in the nucleus (Fig. 3a). Using identical overexpression constructs in telomerase-positive cell lines did not lead to the formation of ZNF524-GFP foci (Supplementary Fig. 2a–c), while other ALT cell lines (GM847, SAOS2, VA13) displayed colocalization of telomeres with ZNF524-GFP (Supplementary Fig. 2d–f). Despite a varying percentage of telomeres with ZNF524-GFP signal among these

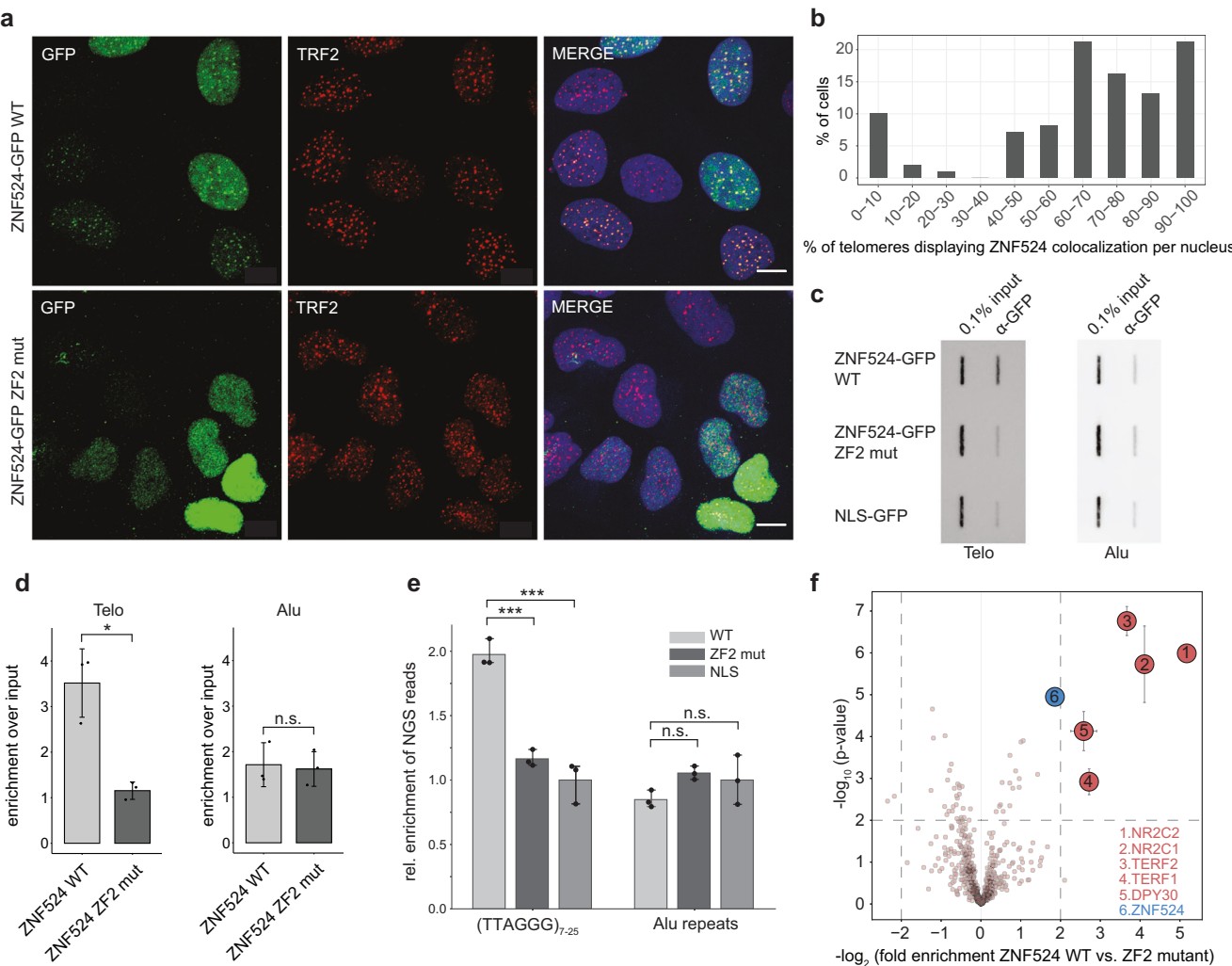

**Fig. 3 | ZNF524 localizes to telomeres in U2OS cells. a** Fluorescence microscopy of colocalization between TRF2 (red) and GFP-ZNF524 (green). Representative images of doxycycline-induced ZNF524-GFP WT and the ZF2 mutant in U2OS cells are shown (scale bar 10 μm). Nuclei were counterstained with DAPI (blue). **b** Quantification of colocalization events of ZNF524-GFP WT with TRF2 ($n = 99$ nuclei). Telomeric foci and the overlap with GFP foci were scored on maximum intensity projections of the acquired z-stacks. **c** Representative slot blot after ChIP using GFP-Trap beads for enrichment of ZNF524-GFP WT and ZF2 mut, visualized using a telomeric probe (left) and an Alu control (right). NLS-GFP served for normalization. **d** Quantification of ChIP experiments normalized to NLS-GFP ($n = 3$ with each replicate seeded and induced independently; mean values are shown with error bars representing SD; *$p < 0.05$ [$p = 0.026$], Welch's test). **e** Quantification of ChIP-seq experiments comparing ZNF524-GFP WT to ZNF524-GFP ZF2 mut and NLS-GFP negative controls. A minimum of 7 and a maximum of 25 hexameric repeats were considered for quantification ($n = 3$ with each replicate seeded and induced independently; The error bars represent 95% confidence intervals; *$p < 0.001$ [WT vs ZF2 mut Telo: $p = 0.00027$; WT vs NLS Telo: $p = 9.66e\text{-}5$; WT vs ZF2 mut Alu: $p = 0.137$; WT vs NLS Alu: $p = 0.283$], one-way ANOVA followed by Dunnett's multiple comparison tests). **f** Volcano plot of BioID assay comparing proximity partners of ZNF524 WT versus ZNF524 ZF2 mut in U2OS cells. BirA*-ZNF524 WT and ZF2 mut were induced with 300 ng ml⁻¹ doxycycline. Specifically-enriched proteins (red numbered circles) are distinguished from background binders by a $> 4$-fold enrichment and $p < 0.01$ (two-sided Student's $t$-test, $n = 4$ with each replicate seeded and induced independently). Two-dimensional error bars represent the standard deviation after iterative imputation cycles during the label-free analysis with substituted zero values (e.g. no detection in the ZF2 mut reaction).

ALT cell lines, it is remarkable that all detectable ZNF524-GFP foci were associated with telomeres. Additionally, we performed chromatin immunoprecipitation (ChIP) with ZNF524-GFP WT compared to either the ZF2 mutant or GFP fused to a nuclear localization signal (NLS-GFP) as negative controls. ZNF524-GFP WT significantly enriched telomeric DNA but not unrelated Alu repeats. Again, the ZNF524-GFP ZF2 mutant failed to enrich telomeric DNA and mirrored the background levels of NLS-GFP (Fig. 3c, d). Notably, while ChIP followed by next-generation sequencing (ChIP-seq) validated the telomeric enrichment (Fig. 3e), we could not detect significant enrichment of ZNF524 to any unique loci in the genome. These data suggest that in contrast to TRF2, RAP1, and ZBTB48, ZNF524 may not act simultaneously as a transcription factor[20,27–31]. This notion is supported by the absence of significant

mRNA expression changes when comparing five U2OS WT and ZNF524 KO clones by RNA sequencing (RNA-seq; see below). To further substantiate the in vivo association with telomeres and to gain insight into the telomeric environment of ZNF524, we performed BioID[32]. We thus expressed a doxycycline-inducible MYC-BirA*-ZNF524 WT or the respective ZF2 mutant in U2OS cells and labeled proximity partners with biotin, followed by streptavidin enrichment and label-free quantitative mass spectrometry analysis[33]. Compared to the ZF2 mutant, ZNF524 WT was in close proximity to known telomere binders in ALT cells: NR2C1, NR2C2, TRF1 and TRF2 (Fig. 3f, Supplementary Data 1). Given that these four proteins are abundantly present at telomeres in U2OS cells, these data further indicate binding of ZNF524 to telomeres in vivo. Since the comparison between ZNF524 WT and the

ZF2 mutant might overlook proteins that are stably associated with ZNF524 independent of telomere binding, we repeated the BioID experiment using MYC-BirA*-NLS as a control. We again enriched the same telomeric proteins but additionally found the ChAHP complex (CHD4-ADNP-CBX3)[34] as well as ZMYM2 and ZMYM3 (Supplementary Fig. 3a, b, Supplementary Data 2). As these proteins were not enriched in comparison to the ZF2 mutant, they are putative ZNF524 interaction partners whose proximity to ZNF524 is not DNA-mediated. Nevertheless, their own association with telomeres might be ZNF524 dependent. Indeed, we confirmed that the association of CBX3 (HP1γ) with telomeres was reduced in the absence of ZNF524 (Supplementary Fig. 3c), suggesting that ZNF524 affects the telomeric chromatin composition.

## TRF2/RAP1 subcomplex is reduced at telomeres upon ZNF524 knockout

To further characterize how ZNF524 may influence telomere function, we created five U2OS and five HeLa ZNF524 knockout (KO) clones by Cas9-mediated genome editing with guide RNAs targeting different regions of the coding sequence along with five WT clones (Supplementary Fig. 4). Loss of ZNF524 protein was confirmed by Western blot, and the presence of frame-shift alterations was verified by next-generation sequencing (Supplementary Fig. 4). As U2OS are ALT-positive cells, we first assessed differences in ALT activity by applying the C-circle assay[35], which quantifies extrachromosomal telomeric DNA commonly occurring during ALT-driven telomere recombination. Amidst high clonal heterogeneity, we could not find a ZNF524-dependent effect on C-circle levels (Supplementary Fig. 5c, d). We then investigated the effect of ZNF524 removal on telomere length in both U2OS and HeLa cells by telomere restriction fragment (TRF) analysis, but we could not consistently observe a significant difference in telomere length in the absence of ZNF524 (Supplementary Fig. 5e–h). This was further supported by qFISH in U2OS WT and ZNF524 KO cells (Supplementary Figs. 4b, 5a). Analysis by flow cytometry also showed that cell cycle dynamics remained stable in the absence of ZNF524 (Supplementary Fig. 5i). However, during validation of our ZNF524 KO clones by immunofluorescence (IF) analysis, we noticed that the TRF2 signal, originally used as a telomeric marker, seemed reduced in the absence of ZNF524. We thus systematically investigated whether ZNF524 influences the localization of shelterin complex members to telomeres. To this end, we performed IF analysis for TRF1, TRF2, RAP1 and POT1 in U2OS WT and ZNF524 KO cells. While the abundance of TRF1 and POT1 at telomeres remained unchanged, we indeed observed a reduction in the fluorescence signal for TRF2 and its interaction partner RAP1 in the ZNF524 KO clones (Fig. 4a–d; Supplementary Fig. 6a, b). This effect was evident both in unsynchronized interphase cells and in native metaphase spreads (Fig. 4a, b, e, f) and was also observed upon ZNF524 knockdown in cells that had not undergone clonal expansion (Supplementary Fig. 6c-e). Importantly, the reduction was rescued in the ZNF524 KO clones by overexpression of ZNF524-HA WT but not by overexpression of the loss-of-binding ZNF524-HA ZF2 mutant (Fig. 4a, b). As expected, overexpression of these constructs did not change TRF1 abundance at telomeres. Of note, the observed signal reduction did not depend on total TRF2 and RAP1 protein levels, as verified by both quantitative Western blot and proteome analysis (Fig. 4g, Supplementary Fig. 7c, Supplementary Data 3). Likewise, the genes of the shelterin complex members were not differentially expressed in ZNF524 KO cells (Supplementary Fig. 7a, b; Supplementary Data 4). Overall, these data suggest that ZNF524 specifically modulates telomeric occupancy of the TRF2/RAP1 subcomplex[36]. This regulation is likely indirect, as we could not detect a physical interaction between ZNF524-GFP and FLAG-TRF2 in coimmunoprecipitation experiments (Supplementary Fig. 7d). When revisiting published protein expression levels of the six shelterin members in 11 cell lines, including U2OS cells[37], we noticed that TRF2

and RAP1 were the most abundant complex members with at least 2-fold higher expression levels than the other shelterin members (Supplementary Fig. 6f). Given the stoichiometry of the shelterin complex[38], the remaining shelterin proteins are concentration-limiting factors for complex formation. Thus, the higher TRF2 and RAP1 abundance potentially results in an excess of this tetrameric sub-complex (two TRF2/RAP1 heterodimers) that was shown to both form in solution and bind to telomeric DNA independently of the shelterin complex[36]. Previous work suggests that each shelterin (sub)complex acts independently of other shelterin complexes in telomere recognition and thus may allow for differences in regulation and functionality[10,36,39,40].

## ZNF524 reduces telomeric DNA damage and recombination

Since TRF2 protects telomeres from the DNA damage response machinery[3,9], we next focused on telomere dysfunction-induced foci (TIFs). Using 53BP1 as a marker, we measured an increase in TIFs in our ZNF524 KO clones for both U2OS and HeLa cells (Fig. 5a, b; Supplementary Fig. 8a, b). To characterize whether the increased DNA damage also results in chromosome aberrations in U2OS, we imaged mitotic telomeres by chromosome orientation fluorescence in situ hybridization (CO-FISH)[41], which specifically stains the parental C- and G-rich telomeric strands, allowing to trace not only fusion events but also telomeric sister chromatid exchanges (t-SCEs). Despite the reduction in TRF2 localization to telomeres in ZNF524 KO clones, we did not observe telomere fusions, the known products of telomere deprotection and subsequent NHEJ. These results are not surprising, as TRF2 is still present at ZNF524 KO telomeres, and minimal levels of TRF2 have been shown to suffice for the prevention of NHEJ[42]. However, we observed an increase in t-SCEs upon ZNF524 knockout (Fig. 5c, d). ALT-positive cells have a basal level of t-SCEs necessary for homologous recombination (HR)-mediated telomere maintenance. Therefore, elevated t-SCE levels are indicative of more recombination events that have been linked to deficiencies in RAP1-mediated HR prevention[15,43]. This aligns with our finding that also telomeric RAP1 is reduced in ZNF524 KO cells (Fig. 4c, d).

## Discussion

Initially only the shelterin complex was described at human telomeres[44], but recently additional direct double-stranded telomeric proteins have been discovered. Beyond HOT1, which, like TRF1 and TRF2, binds telomeres via a homeobox domain[19], the discoveries of ZBTB48 and ZBTB10 identified zinc fingers as another protein domain to recognize telomeric repeats[24,25]. Here, we report that ZNF524, a protein of previously unknown function, binds to telomeric DNA in vitro and localizes to telomeres in vivo. Furthermore, it directly interacts with telomeric DNA, as shown in our 2.40 Å crystal structure, and we propose that the RxxHxxR motif is a recurring theme for zinc finger proteins binding to TTAGGG repeats. Colocalization of ZNF524-GFP to telomeres was observed in several ALT cells, demonstrating ZNF524's ability to bind telomeres in vivo. While the increase in TIFs in both ALT (U2OS) and telomerase-positive (HeLa) cells suggests that the function of ZNF524 is not restricted to ALT telomeres, the absence of telomeric ZNF524 foci in telomerase-positive cells implies that ZNF524 might act predominantly as a telomere-binding protein in ALT cells.

The phenotypes observed in the absence of ZNF524 are particularly intriguing given that they entail reduced TRF2 and RAP1 at telomeres. Both proteins form a previously described binary complex[10,36] for which we here suggest independent in vivo regulation compared to the fully assembled shelterin complex (Fig. 5e). Both the increase in TIFs and the increase in t-SCEs imply an involvement of ZNF524 in telomere protection through regulation of the TRF2/RAP1 subcomplex. As our findings indicate that ZNF524 neither regulates TRF2/RAP1 expression nor directly interacts with TRF2, it

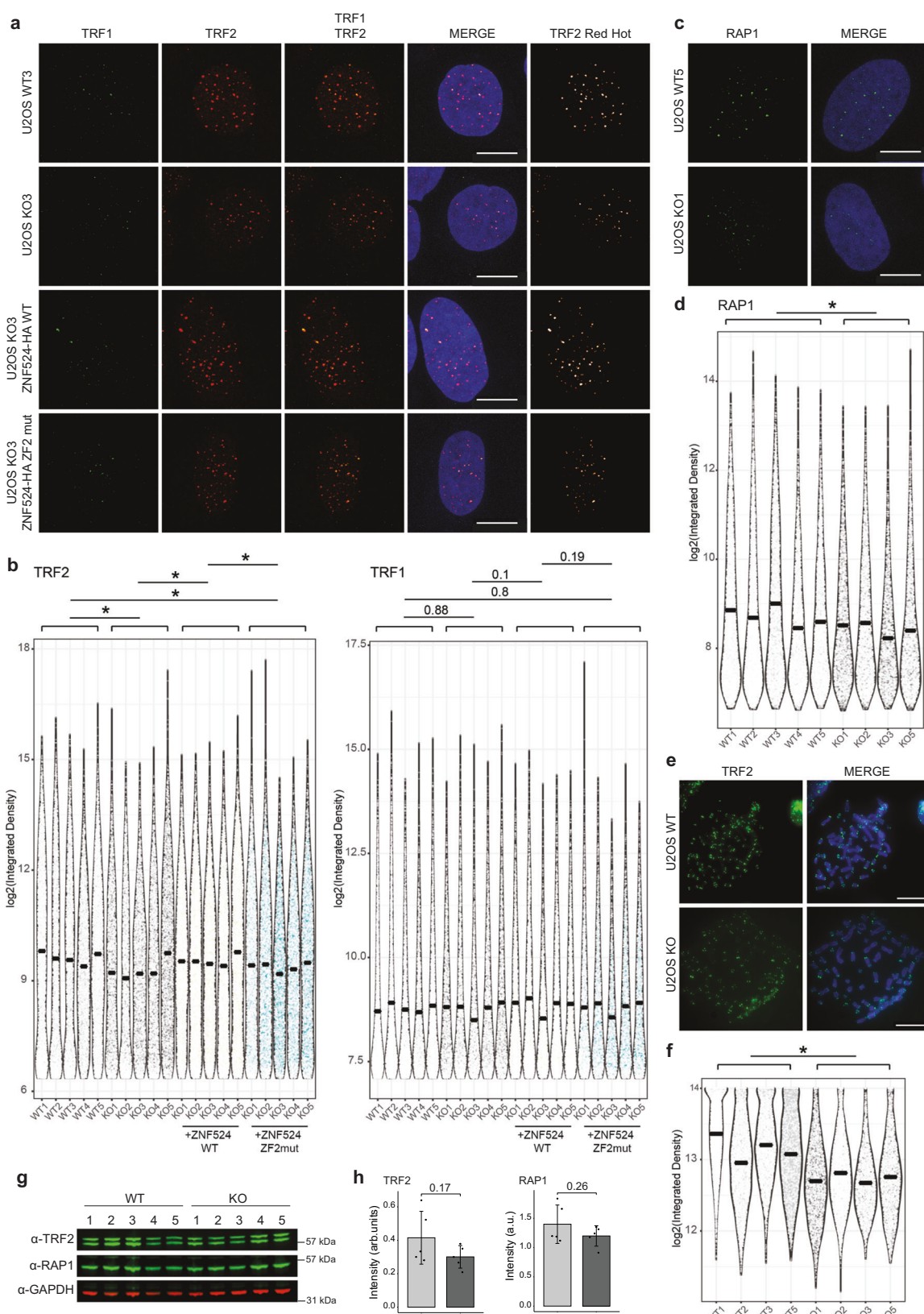

remains to be determined how ZNF524 influences TRF2/RAP1 locali-
zation to telomeres. One appealing mechanism may involve ZNF524-
dependent changes in epigenetic marks at telomeres, similar to the
ZNF827-mediated recruitment of the NuRD complex to telomeres[45].
This notion is supported by the ZNF524-dependent telomere associa-
tion of HP1γ, as a representative member of the ChAHP complex,

suggesting an altered chromatin status when ZNF524 is lacking. More
generally, this could even extend to the interplay between telomeres
and the nuclear lamina. Indeed, it was recently reported that lamin B1
overexpression led to a partial displacement of TRF2/RAP1 and a
concomitant increase in telomeric instability[46], reminiscent of the
ZNF524 KO phenotype. Alternatively, ZNF524 might be involved in the

**Fig. 4 | ZNF524 KO leads to a reduction in TRF2 and RAP1 at telomeres.**
**a** Representative immunofluorescence images of U2OS WT and ZNF524 KO cells stained for TRF1 (green) and TRF2 (red). Nuclei were counterstained with DAPI (blue). **b** Quantification of TRF1 and TRF2 IF signals in WT and KO clones as well as KO clones induced with 2 µg mL$^{-1}$ doxycycline for expression of HA-ZNF524 WT or HA-ZNF524 ZF2mut. The violin plot shows the individual data points as densities. A total of 1487-4576 telomeres per clone were analyzed for TRF1, and 2300-5290 telomeres per clone for TRF2; *$p < 0.05$ ($n = 5$). For TRF2 IF signal quantification, the $p$-values are as follows: WT vs KO: $p = 0.0253$; WT vs KO + ZNF524 WT: $p = 0.220$; WT vs KO + ZNF524 ZF2 mut: $p = 0.0131$; KO vs KO + ZNF524 ZF2 mut: $p = 0.760$; KO vs KO + ZNF524 WT: $p = 0.011$; KO + ZNF524 WT vs KO + ZNF524 ZF2 mut: $p = 0.0112$ (**c**) Representative IF pictures of U2OS WT and ZNF524 KO cells stained for RAP1 (green). **d** Quantification of the RAP1 IF signal in WT and KO clones depicted as

violin plots. 2245-5290 telomeres per clone were analyzed; *$p < 0.05$ [$p = 0.0249$] ($n = 5$). **e** Representative IF pictures of native metaphase spreads of U2OS WT and ZNF524 KO clones stained for TRF2 (green). **f** Quantification of the TRF2 signal on metaphase spreads depicted as violin plots. A total of 1494-6875 telomeres per clone were analyzed; *$p < 0.05$ [$p = 0.013$] ($n = 4$). **g** Quantitative Western blot showing total TRF2 and RAP1 protein levels in U2OS WT and ZNF524 KO clones with GAPDH as a loading control. **h** Quantification of TRF2 and RAP1 signals normalized to GAPDH. The bar plot shows the mean intensities ± SD. The intensity values of the individual clones are depicted as black dots. Statistical comparison by a two-sided Student's $t$-test ($n = 5$). For all data n represent the number of independent clones used as biological replicates. For all IF data, scale bars represent 10 µm in all panels; $p$-values were determined by a paired one-sided Student's $t$-test; the mean is indicated by a solid line.

formation of different structures at telomeres that in turn affect TRF2 binding. For example, TRF2 is known to bind not only telomeric dsDNA but also junctions. In vitro experiments showed that TRF2 indeed preferred ds-/ss- junctions over dsDNA[47] and recognized 3- and 4-way junctions. These structures are found at the base of loops, in replication forks and in recombination intermediates, respectively. TRF2 aids in the formation of these structures while also stabilizing them via the N-terminal basic domain[48,49].

One might speculate how the absence of ZNF524 would lead to the observed increase in telomere damage and recombination events. As mentioned above, epigenetic changes could lead to impaired telomere integrity, for example by changing telomere accessibility not only for TRF2 but also for other transient factors. Redundancies with epigenetic factors or factors auxiliary to chromatin remodelers, like ZNF827 to the NuRD complex, could explain why ZNF524-specific phenotypes seem subtle in unchallenged cells. Future investigations of the telomeric chromatin state in the presence and absence of ZNF524 could therefore further reveal the mechanism by which ZNF524 influences TRF2/RAP1 localization and how it safeguards telomeres from unscheduled recombination and DNA damage signaling. In such a model, ZNF524 might recruit the ChAHP complex to telomeres to maintain a protective chromatin environment. In return, ZNF524 might regulate the ALT-promoting activity of the NuRD complex by sequestering the mutual complex member CHD4 together with HP1γ in a competition for interactions[50]. Similarly, ZNF524-ChAHP abundance at telomeres may balance the action of ADNP's counterpart POGZ, a protein that promotes homology-directed DNA repair in an HP1-dependent manner and that was recently found to associate with telomeric chromatin in ALT cells[51–53]. Additionally, the other putative ZNF524 interaction partners ZMYM2 and ZMYM3 have been shown to restrict 53BP1 loading at double-strand breaks[54], providing potential links between ZNF524-dependent epigenetic changes and telomere integrity.

Alternatively, the described telomere instability could be directly linked to the ZNF524-dependent abundance of TRF2/RAP1 at the telomeres. The reduction (but not complete removal) of TRF2/RAP1 at telomeres in the absence of ZNF524 is reminiscent of a previously described intermediate state of telomere deprotection, where the integrity of telomeres is compromised but not to the point of detrimental chromosome rearrangements. This intermediate state often coincides with TRF2 reduction, for example caused by prolonged mitotic arrest, oxidative stress or partial knockdown of TRF2[42,55–57]. As a result, the DNA damage markers 53BP1 and γH2AX are recruited to telomeres, and an increase in t-SCEs has also been reported[42,55–57]. These effects were recapitulated in our ZNF524 KO cells. It is therefore possible that the phenotypes observed here are a result of TRF2/RAP1 reduction at telomeres, although TRF2-independent effects of ZNF524 removal cannot be excluded at this point. Also matching our findings, intermediate state telomeres lack telomere fusions, and amplified polyploidy does not occur. Compared to cells undergoing telomeric NHEJ, the activation of ATM followed by CHK2 phosphorylation is less

pronounced in the intermediate state, which could explain why we did not detect a global upregulation of pATM and pCHK2 despite an increase in TIFs (Supplementary Fig. 8c–e; Fig. 5a, b). Overall, the disruption of ZNF524 seems to evoke an intermediate state of telomere deprotection.

Notably, each of the recently identified proteins (ZNF524, ZBTB48, ZBTB10, and HOT1) is less abundant at telomeres than shelterin, and their precise molecular mechanisms are only beginning to surface. However, our data suggests a picture of telomeres with a heterogeneous composition, including the entire shelterin complex, its potential subcomplexes and several other direct telomere binding proteins (Fig. 5e). With the emergence of this larger network, the interplay between them will be paramount in understanding telomere biology, as changes in telomeric phenotypes will need to be interpreted in the context of their complete network.

## Methods
### Cell culture
U2OS, HeLa (Kyoto), HeLa 1.3[9], HT1080ST[58], GM847, Saos2, and WI-38 VA-13 cells were cultivated in 4.5 g L$^{-1}$ Dulbecco's modified eagle medium (DMEM) supplemented with 10% fetal bovine serum (Gibco), 2 mM glutamine (Thermo Scientific), 100 U mL$^{-1}$ penicillin and 100 µg mL$^{-1}$ streptomycin (Gibco). Cells were kept at 37 °C and 5% CO$_2$ in a humidified incubator.

### Cloning and plasmids
ZNF524 was obtained from the Orfeome collection (Q96C55; ENSG00000171443). Zinc finger mutations were introduced by site-directed mutagenesis using specific primers (see Supplementary Data 5), and the sequences were confirmed by Sanger sequencing (GATC). The constructs were LR-recombined into pDest-pcDNA3.1 with an N-terminal FLAG-tag or into pLIX_403 (Plasmid #41395, Addgene) with a C-terminal GFP-tag. pLIX_403 was a gift from David Root (Addgene plasmid #41395; http://n2t.net/addgene:41395; RRID:Addgene_41395). pTRIPZ carrying MYC-BirA*[59] was modified for 3rd generation lentivirus production and digested with Xho I and Mlu I. The insert was PCR amplified to introduce the respective overhangs and ligated into the vector backbone. Sequences were confirmed by Sanger sequencing (1st BASE). For ZNF524 rescue, sequential site-directed mutagenesis was used to introduce silent point mutations in ZNF524 pCR8/GW/TOPO, resulting in a sequence that is resistant to the sgRNAs used for the CRISPR/Cas9 KO clones (see Supplementary Data 5). For the ZF2 mut, an additional C144A mutation was included. The rescue construct was introduced into pInducer20 by LR recombination. pInducer20 was a gift from Stephen Elledge (Addgene plasmid # 44012; http://n2t.net/addgene:44012; RRID:Addgene_44012)[60].

### Transfection
Plasmids were transfected into HeLa Kyoto and HEK293 cells using linear polyethylenimine (PEI, MW 25,000; Polysciences). One day prior to transfection, cells were seeded in a 10 cm cell culture dish with

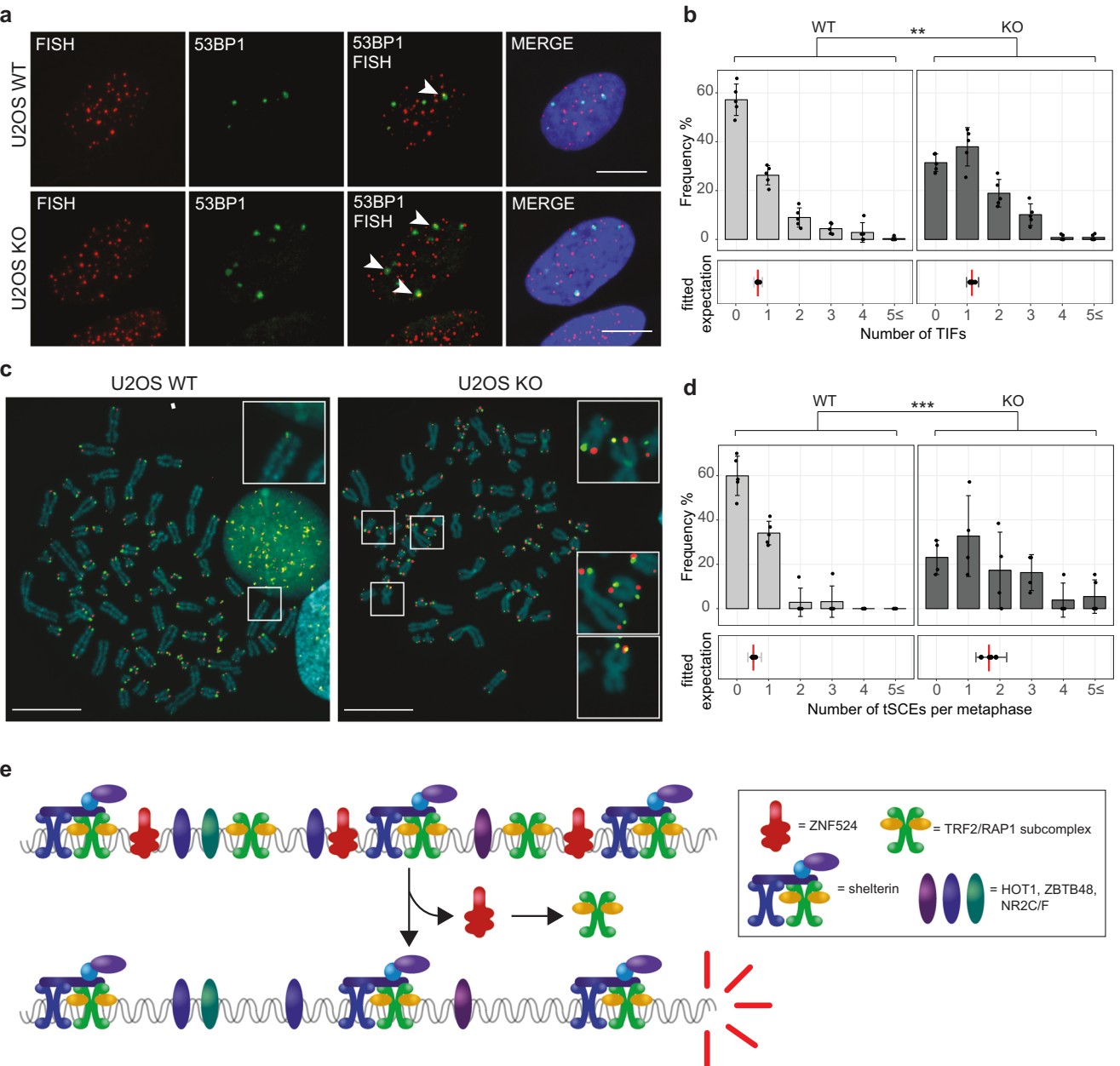

**Fig. 5 | Removal of ZNF524 results in telomeric aberrations. a** 53BP1 immuno-fluorescence staining (green) coupled with telomeric FISH (red) indicates telomere dysfunction-induced foci (TIFs, white arrows), scale bars represent 10 μm. Nuclei were counterstained with DAPI (blue). **b** Quantification of TIFs per cell; 5 WT and 5 KO clones were counted with at least 35 nuclei per clone; upper plot: Frequency of cells with the indicated number of TIFs; error bars represent SD; lower plot: the vertical lines (red) represent the fitted expected number of TIFs (GLMM for nega-tive binomially distributed data). Error bars represent 95% confidence intervals for the mean number of TIFs. The *p*-value was calculated using a likelihood ratio test; **\*\****p* < 0.01 (*n* = 5 based on 5 independent clones for all conditions). **c** CO-FISH with Cy3-labeled G-rich telomere probe (red) and FITC-labeled C-rich telomere probe (green). Scale bars represent 10 μM. Metaphases were counterstained with DAPI (blue). **d** Quantification of telomeric sister chromatid exchanges (t-SCE) per metaphase; 5 WT and 4 KO clones were counted with at least 10 metaphase spreads per clone; upper plot: Frequency of cells with the indicated number of t-SCEs; error bars represent SD; lower plot: the vertical lines (red) represent the fitted expected number of t-SCEs (GLMM for negative binomially distributed data). Error bars represent 95% confidence intervals for the mean number of t-SCEs. The *p*-value was calculated using a likelihood ratio test; \*\*\**p* < 0.001 (*n* = 5 for WT and *n* = 4 for KO). **e** Schematic model of ZNF524's proposed function at telomeres.

450,000 cells mL⁻¹. For transfection, 48 μL PEI and 12 μg plasmid were diluted in DMEM and added to the cells. The medium was replaced after 6-8 h. Plasmid transfection in U2OS cells was performed using the Amaxa Cell Line Nucleofector Kit V (Lonza) according to the manu-facturer's instructions.

**Cloning and protein expression**
For truncated ZNF524 ZF1-4 proteins (including 107-237 aa for crys-tallization and 110-223 aa for ITC experiments), their corresponding

coding sequences were amplified by PCR from a human brain cDNA library and inserted into a modified pGEX-4T-1 vector with a Tobacco Etch Virus (TEV) cleavage site instead of the original thrombin cleavage site. The recombinant plasmids were transformed into *Escherichia coli* BL21(DE3) cells, and then the cells were cultured in LB medium sup-plemented with 100 mM ZnCl$_2$ at 37 °C. When the culture reached an A$_{600\ nm}$ O.D. of ~1.0, protein expression was induced with 0.2 mM isopropyl b-D-1-thiogalactopyranoside (IPTG, Thermo Scientific) at 16 °C for 24 h. The cells were then collected by centrifugation and lysed

in buffer containing 20 mM Tris-HCl (pH 7.5) and 1 M NaCl. Next, cell debris was removed by centrifugation at 22,000 × g for 30 min. After centrifugation, the GST-tagged protein was purified by glutathione sepharose (GE Healthcare), followed by on-column TEV cleavage. The tag-free protein was further purified by size-exclusion chromatography on a Hiload 16/60 Superdex 75 column (GE Healthcare). Mutants and ZF truncations were expressed and purified using the same procedure as for the wild-type proteins. All purified proteins were changed to a buffer containing 20 mM Tris-HCl (pH 7.5) and 150 mM NaCl for the following experiments.

### ITC

All ITC experiments were conducted using a MicroCal PEAQ-ITC instrument (Malvern Panalytical) at 20 °C. The protein and dsDNA samples were prepared by buffer exchange to a buffer containing 20 mM Tris-HCl (pH 7.5) and 150 mM NaCl. Thermodynamic analysis was performed by titrations of dsDNA (syringe samples) into cell protein, with an initial injection of 1 μL followed by 19 consecutive injections of 2 μL, each separated by a time interval of 120 s. The titration protocol consisted of a single initial injection of 1 μl, followed by 19 injections of 2 μl dsDNA samples into the sample cell containing ZNF524 proteins. Thermodynamic data were analyzed with a single-site binding model using MicroCal PEAQ-ITC Analysis Software provided by the manufacturer. The binding isotherms were integrated to give the enthalpy change ΔH plotted as a function of the molar ratio of DNA and protein. The initial titration point was always discarded.

### Crystallization

ZNF524 ZF1-4 protein (107-237 aa) was mixed with double-stranded telomeric oligonucleotide (G-strand: 5′-AGGGTTAGGGTTAGG-3′; C-strand: 5′-TCCTAACCCTAACCC-3′) at a 1:1.2 molar ratio. The mixture was dialyzed against a buffer containing 20 mM Tris-HCl (pH 7.5) and 150 mM NaCl overnight at 4 °C. After dialysis, the protein-DNA complex was further purified using a Hiload 16/60 Superdex 75 column (GE Healthcare). Then, the complex was concentrated to ~15 mg mL⁻¹. The ZNF524-dsDNA complex crystal was grown at 293 K by using the hanging drop vapor diffusion method with the mother liquor, 0.2 M imidazole malate, pH 5.5 and 15% (w/v) polyethylene glycol 600.

The X-ray diffraction dataset was collected on beamline 19U1 of the Shanghai Synchrotron Radiation Facility (SSRF). The diffraction data were processed using *HKL2000* software (HKL Research)[61]. Experimental phasing was performed with the *AutoSol* program in the PHENIX suite, with the zinc atom sites being found by the *SHELX C/D* program[62,63]. The *Buccaneer* program in the CCP4 suite was utilized for model building, and further manual building and refinements of the protein and the DNA duplex were completed with *COOT* and *Phenix.refine*, respectively[62,64,65]. Crystal diffraction data and refinement statistics are shown in Supplementary Table 3.

### Lentiviral transduction

HEK293T cells were seeded to a confluence of 70% in DMEM without antibiotics. They were transfected with the vectors pMDLg/pRRE, pRSV-Rev, pMD2. G and the respective expression vector (pLIX403-GFP for ZNF524-GFP constructs, pTRIPZ for BirA*-ZNF524 constructs and pInducer20 for ZNF524-HA constructs) using linear polyethylenimine (PEI MW 25,000, Polysciences Inc.), and the medium was exchanged after 24 h. After another 24 h, the supernatant containing the virus was collected, filtered at 0.45 μm and supplemented with 8 μg mL⁻¹ polybrene and 10 mM HEPES buffer pH 7.5. Target U2OS cells were seeded to 50% confluence. The medium was replaced with virus-containing supernatant, incubated for 24 h and then exchanged for fresh medium. Cells were selected with 2 μg mL⁻¹ puromycin for 48 h after transfection and subsequently cultured in the presence of 1 μg mL⁻¹ puromycin. Cells transfected with pInducer20 were selected

with 400 μg mL⁻¹ G418 for 6 days and subsequently cultured in the presence of 100 μg mL⁻¹ G418.

### Western blots

Protein samples were size-separated in a 4-12% Bis-/Tris gel (NuPAGE, Thermo Scientific) run in 1x MES buffer (Thermo Scientific) at 180 V for 45 min (70 min for pATM). Denatured proteins were then transferred to a nitrocellulose membrane (Amersham Protran, VWR) by applying 300 mA for at least 60 min in a wet transfer chamber with blotting buffer (25 mM Tris pH 8.3, 192 mM glycine, 20% MeOH). The membrane was first incubated with blocking buffer (5% (w/v) skim milk) at RT for 1 h and then with the respective primary antibody diluted in blocking buffer. Information about the antibodies can be found in Supplementary Data 6. The corresponding secondary antibody (Supplementary Data 6) was added at a 1:3,000 dilution in blocking buffer for 1 h at RT followed by PBS-T washes. Bands were detected by Super Signal West Pico Chemiluminescent Substrate (Thermo Scientific) according to the manufacturer's instructions using a ChemiDoc Imaging Systems (Bio-Rad). The Penta-His HRP Conjugate Kit (Qiagen) was used for detection of His-tagged proteins according to the manufacturer's instructions. For quantitative Western blots, the membrane was blocked with 5% BSA in Tris-buffered saline (TBS) for 1 h at RT and subsequently incubated with the respective antibody (see Supplementary Data 6) in 5% BSA TBS-T. After washing with TBS-T, the corresponding fluorescently labeled antibody was added in a 1:15000 dilution in 5% BSA TBS-T and incubated for 1 h at RT in the dark. The membrane was washed in TBS-T, and the bands were detected using the LI-COR Odyssey (LI-COR).

### In vitro reconstitution DNA pull-downs

Biotinylated DNA for the DNA pull-down was prepared as previously described[16,19,66]. To prepare the biotinylated bait DNA, 25 μL of the 10-mer telomeric repeat oligonucleotide or the scrambled control sequence oligonucleotide was mixed with 25 μL of their reverse complement counterparts. After the addition of 10 μL of annealing buffer (200 mM Tris-HCl pH 7.5, 100 mM MgCl₂, 1 M KCl), the reaction was brought to a final volume of 100 μL with ultrapure water and heated at 80 °C for 5 min. Subsequently, the reaction was slowly cooled to RT by switching off the Eppendorf thermomixer and letting it cool. For phosphorylation, 55 μL ultrapure water, 20 μL 10x T4 DNA ligase buffer (Thermo Scientific), 10 μL PEG 6000, 10 μL 100 mM ATP, 2 μL 1 M DTT and 27.5 μL of T4 polynucleotide kinase (Thermo Scientific) were added to the annealed oligonucleotides, and the reaction was incubated for 2 h at 37 °C. For overnight ligation at RT, the reaction was treated with 4 μL of T4 DNA ligase (Thermo Scientific). Successful oligomerisation was confirmed by agarose gel electrophoresis. For phenol/chloroform extraction, 1 volume of ultrapure water and 200 μL phenol/chloroform/IAA (25:24:1) pH 8.0 (Thermo Scientific) were added to the DNA followed by mixing and 2 min centrifugation at 16,000 g. The aqueous phase was transferred to a new Eppendorf tube and mixed with 1 mL 100% EtOH. Precipitation for 30 min at −20 °C was followed by 45 min centrifugation at 16,000 g and 4 °C. The pellet was taken up in 74 μL ultrapure water and biotinylated at 37 °C overnight through the addition of 10 μL 10x polymerase buffer (reaction buffer for Klenow fragment), 10 μL 0.4 mM biotin-7-dATP (Jena Bioscience, NU-835) and 6 μL DNA polymerase 30 units (Klenow fragment exo-5 U μL⁻¹). Finally, the biotinylated bait oligonucleotides were purified using Microspin Sephadex G-50 columns according to the manufacturer's instructions (GE Healthcare).

The pull-down with biotinylated bait DNA was performed as follows: per sample, 20 μL biotinylated oligo bait were diluted in 200 μL PBB buffer (150 mM NaCl, 50 mM Tris-HCl pH 7.5, 0.5% IGEPAL CA-630, 5 mM MgCl₂, 1 mM DTT, 1x protease inhibitor EDTA free [Roche]) and mixed with 50 μL PBB-buffer-equilibrated MyOne Streptavidin C1 Dynabeads (Thermo Scientific). After 15 min incubation at RT, the

DNA-coupled beads were washed three times with PBB buffer. Subsequently, the beads were taken up in 150 μL PBB buffer supplemented with 15 μg salmon sperm DNA. Then, 400 μg of cell lysate was added to the DNA bait-coupled beads and incubated for 90 min at 4 °C on a rotating wheel followed by three washes with 500 μL of PBB buffer. The proteins were eluted from the beads by 25 μL of 1x NuPAGE LDS sample buffer supplemented with 100 mM DTT (Sigma–Aldrich) and heated for 10 min at 70 °C. The beads were again separated using a magnetic rack, and the entire volume was then loaded onto a precast 10-well 4-12% Bis-/Tris NuPAGE gel to separate the proteins for 45 min at 180 V in 1x MES buffer (Thermo Scientific).

## Chromatin immunoprecipitation

U2OS stable cell lines carrying ZNF524-GFP WT, ZNF524-GFP ZF2 mut or NLS-GFP were seeded in medium supplemented with 300 ng mL$^{-1}$ doxycycline 48 h prior to the experiment to induce expression. The attached cells were washed with ice-cold PBS twice and then crosslinked with 1% (v/v) formaldehyde without methanol (Thermo Scientific) in DMEM for exactly 20 min at RT. The reaction was quenched with 2.5 M glycine in PBS for 5 min at RT. Subsequently, the entire medium was removed, and the cells were washed twice with cold PBS and then scraped from the cell culture dishes in 1 mL PBS with cOmplete protease inhibitor by Roche. The cells were washed once with lysis buffer 1 (140 mM NaCl, 50 mM Tris-HCl pH 8.0, 250 mM sucrose, 1 mM EDTA, 10% glycerol, 0.5% IGEPAL CA-630, 0.25% Triton X-100, 0.25% TWEEN 20, cOmplete protease inhibitor by Roche) for 15 min at 4 °C and centrifuged at 1,000 g at 4 °C for 5 min. The supernatant was removed, and the cell pellet was washed in lysis buffer 2 (200 mM NaCl, 10 mM Tris-HCl pH 8.0, 1 mM EDTA, 0.5 mM EGTA, cOmplete protease inhibitor by Roche). Cells were taken up in sonication buffer (50 mM Tris-HCl pH 8.0, 10 mM EDTA, 1% SDS, cOmplete protease inhibitor by Roche) at a ratio of 10 million cells in 150 μL (120 μL for ChIP-seq). During sonication, the cells were kept on ice. To obtain chromatin fragments of ~200–500 bp, the following settings on the EpiShear probe sonicator (Active Motif) were used: Amplitude of 30%, 15 s ON and 30 s OFF, 25 cycles. After sonication, the suspension was centrifuged at 20,000 g at 4 °C for 10 min, and the supernatant was retained. To verify successful sonication, 10 μL of sonicate was mixed with 200 mM NaCl and 1 mg mL$^{-1}$ RNase A to a final volume of 100 μL and incubated at 37 °C for 1 h. The mixture was supplemented with 0.4 μg mL$^{-1}$ proteinase K and then incubated at 62 °C for 2 h. After purification using the Qiagen PCR purification kit according to the manufacturer's instructions, the chromatin fragment size was verified on a 1.5% agarose gel. For immunoprecipitation, 36.5 μg chromatin (100 μg for ChIP-seq) was mixed with 5 volumes of modified PBB buffer (180 mM NaCl, 50 mM Tris-HCl pH 8.0, 0.25% IGEPAL CA-630, 1 mM DTT, 5 mM MgCl$_2$, cOmplete protease inhibitor by Roche). Per replicate and construct, 8 μL (35 μL for ChIP-seq) of GFP-Trap magnetic agarose beads (Chromotek) were equilibrated in PBB buffer, blocked in PBB buffer supplemented with 10 μg mL$^{-1}$ BSA and sheared salmon sperm DNA (Thermo Scientific) and finally taken up in PBB buffer. The blocked and equilibrated beads were added to the chromatin and incubated at 4 °C on a rotating wheel overnight. After at least 16 h of incubation, the immunoprecipitate was washed with PBB buffer (150 mM NaCl) 5 times. Finally, the beads were washed once with TE buffer (10 mM Tris, 1 mM EDTA), and the supernatant was removed completely. The beads were suspended in filtered elution buffer (1% (v/v) SDS, 0.1 M NaHCO$_3$) and incubated at 60 °C for 30 min. The supernatant containing the chromatin was kept, and the elution step was repeated once. To reverse the crosslinking, the elution was supplemented with NaCl to a final concentration of 200 mM and incubated at 65 °C overnight. On the following day, 60 μg RNase A was added, and the mixture was incubated at 37 °C for 30 min. Finally, 0.01 mM EDTA, 20 mM Tris-HCl pH 6.5 and 2.5 μg mL$^{-1}$ proteinase K were added and incubated at 45 °C for 2 h. The chromatin was then purified using the QIAquick PCR purification Kit (Qiagen) according to the manufacturer's instructions, and the DNA was eluted in TE buffer.

## Slot blot

For detection of C-circles, 6 μL of the reaction was diluted to 100 μL in 2x SSC (3 M NaCl, 0.3 M sodium citrate) and slot-blotted on a Hybond XL nylon membrane (GE Healthcare). For detection of ChIP samples, the eluted chromatin was denatured at 95 °C. Subsequently, 10 μL of denatured chromatin was slot-blotted and hybridized with either telomeric ((CCCTAA)$_4$) or Alu (TGGCTCACGCCTGTAATCCCAGC ACTTTGGGAGGCCGA) DIG-labeled probes. The TeloTAGGG Telomere Length Assay kit (Sigma–Aldrich) was used according to the manufacturer's instructions. After blotting, the membrane was UV cross-linked at 120 mJ using a Stratalinker® UV Crosslinker (Stratagene) and rinsed with HPLC water and twice with 2x SSC prior to incubation in prewarmed DIG Easy Hyb Granules for 60 min at 42 °C with gentle agitation for prehybridization. The DIG-labeled probe (telomere or Alu), diluted 1:5000 in Hyb Granules, was added for hybridization and incubated for 3 h or overnight at 42 °C with gentle agitation. The membrane was washed twice with stringent wash buffer I (2x SSC, 0.1% SDS) for 5 min at RT followed by two washes with prewarmed stringent wash buffer II (0.2x SSC, 0.1% SDS) for 15 min at 50 °C and a wash in 1x washing buffer for 5 min at RT. Next, the membrane was incubated in 1x blocking solution for 30 min followed by incubation with anti-DIG-AP antibody (1:10,000) diluted in 1x blocking solution for either 30 min at RT or overnight at 4 °C. Following two washes with 1x washing buffer for 15 min each, the membrane was incubated with 1x detection buffer for 5 min. For detection of the samples, CDP-star substrate solution was added to the membrane before visualization using either X-ray films or a ChemiDoc Touch Imaging System (Bio-Rad). Quantification of the signal intensity was performed using Fiji (ImageJ). The background was subtracted, and an equal area was measured for each slot. The integrated density was subjected to Student's $t$-test analysis.

## Next-generation chromatin immunoprecipitation sequencing (ChIP-seq)

ChIP reactions were prepared as described above using 100 μg chromatin as starting material. The purified DNA fragments were submitted to NovogeneAIT for ChIP-seq sample preparation and sequencing. In brief, the DNA fragments were repaired, A-tailed and then ligated with Illumina adapters. After size selection and PCR amplification, the sequencing library was checked for size distribution using the 2100 Bioanalyzer System (Agilent) and quantified using real-time PCR and the Qubit dsDNA HS Assay kit on a Qubit 2.0 fluorometer (Thermo Scientific). The quantified libraries were pooled in equimolar ratios and sequenced on a NovaSeq 6000 (Illumina).

For each sample, 39 to 52 million reads were obtained as 150 bp paired-end reads. The reads were mapped to the human reference genome version GRCh38 using Bowtie 2 version 2.3.5.1 with default settings and processed using SAMtools version 1.12[67,68]. Unique alignments were obtained by filtering alignments having a MAPQ score of 40 or more using samtools version 1.12. Bigwig tracks normalized to counts per million mapped reads were produced using deeptools 3.5.0, and peaks were called using MACS version 2.2.7.1 in paired-end mode with the default q-value cutoff of 0.05[69,70].

## Nuclear protein extraction

Cells were harvested and washed once in PBS. The following steps were performed on ice or at 4 °C. Five volumes of cold buffer A (10 mM HEPES-KOH pH 7.9, 1.5 mM MgCl$_2$, 10 mM KCl) were added to the pellet and incubated on ice for 10 min. After centrifugation for 5 min at 450 g, the cells were suspended in 2 pellet volumes of cold buffer A+ (buffer A, 0.1% IGEPAL CA-630, 1x complete protease inhibitor cocktail by Roche) and lysed using a glass dounce homogenizer. The lysate was centrifuged for 15 min at 1,500 g, and the pellet was washed with 10

volumes of PBS and suspended in 2 volumes of cold buffer C+ (420 mM NaCl, 20 mM HEPES·KOH pH 7.9, 20% glycerol, 2 mM MgCl₂, 0.2 mM EDTA pH 8.0, 0.1% IGEPAL CA-630, 0.5 mM DTT, 1x complete protease inhibitor cocktail by Roche). After a 1 h incubation at 4 °C, the chromatin was separated from the soluble nuclear fraction by a 1 h centrifugation at 20,000 g at 4 °C. The protein concentration of the nuclear fraction was determined by Bradford or BCA assay before snap freezing it in liquid nitrogen and storing it at −80 °C.

## BioID

U2OS cell lines carrying the BirA*-ZNF524 WT or BirA*-ZNF524 ZF2 mutant were induced with 100 ng mL⁻¹ doxycycline 48 h prior to harvest. For the comparison between ZNF524 WT and NLS, BirA*-ZNF524 WT was induced with 300 ng mL⁻¹ and BirA*-NLS with 40 ng mL⁻¹ doxycycline 48 h prior to harvest. After 42 h, the cells were additionally treated with 50 µM biotin for 6 h to allow for the biotinylation of proteins proximal to the target protein. Subsequently, nuclear extract was prepared as described above. Biotinylated proteins were isolated from the extract using MyOne Streptavidin C1 Dynabeads (Thermo Scientific). To this end, 150 µL Dynabeads were washed with PBB+ buffer (420 mM NaCl, 50 mM Tris-HCl pH 8.0, 5 mM MgCl₂, 0.25% IGEPAL CA-630, 1 mM DTT, cOmplete protease inhibitor [Roche]) and mixed with 200 µL of 3 µg µL⁻¹ nuclear extract and 500 µL PBB+ buffer. After 2 h incubation at 4 °C on a rotating wheel, the beads were pelleted on a magnetic rack and washed thrice with ice-cold PBB+ buffer. Finally, the isolated proteins were eluted in 25 µL of 2x Laemmli buffer (Sigma Aldrich) by boiling for 5 min at 95 °C, and the supernatant was prepared for mass spectrometry measurement.

## MS sample preparation

In-gel digestion was performed as previously described[16,19,66,71]. Essentially, denatured proteins were separated on a 4-12% Bis-/Tris gel (NuPAGE, Thermo Scientific) for 10 min (30 min for proteome) at 180 V in 1x MOPS buffer (NuPAGE, Thermo Scientific). Proteins were stained with the Colloidal Blue Kit (Thermo Scientific) according to the manufacturer's instructions or fixed (7% acetic acid, 40% methanol) for 15 min and then stained with Coomassie Blue (Roth). For the proteome, each sample was divided into 4 fractions according to molecular weight. Each sample was cut individually with a clean scalpel into 1 mm×1 mm pieces. The gel pieces were destained (50% 50 mM NH₄HCO₃ (ABC), 50% ethanol) in several rounds, dehydrated twice in 100% acetonitrile, and its remnants were removed using a Concentrator Plus (Eppendorf). The gel pieces were incubated with reduction buffer (50 mM ABC pH 8.0, 10 mM DTT) for 60 min at 56 °C followed by incubation with alkylation buffer (50 mM ABC pH 8.0, 50 mM iodoacetamide) for 45 min at RT in the dark. The gel pieces were washed once with 50 mM ABC. Subsequently, two dehydration steps were performed, and the acetonitrile was completely removed. The gel pieces were rehydrated in trypsin solution (50 mM ABC pH 8.0, 1 or 2 µg trypsin per sample for proteome or BioID, respectively, Promega) overnight at 37 °C. The supernatant was collected, and the digested peptides were extracted by one round of incubation with extraction buffer (30% acetonitrile, 10% trifluoroacetic acid), one round of 100% acetonitrile, another round of extraction buffer and finally two rounds of 100% acetonitrile for 15 min at RT with agitation. In each round, supernatants were recovered and combined. The acetonitrile was evaporated in a Concentrator Plus to reduce the volume to <200 µL.

Stage tip purification of the samples was performed as previously described[72]. Therefore, 2 layers of Empore C18 material (3 M) were stacked in a 200 µL pipet tip, and the material was activated with methanol, equilibrated with mixture B (80% acetonitrile, 0.5% formic acid/0.1% formic acid for proteome) and then washed with mixture A (0.5% formic acid/0.1% formic acid for proteome). After applying the sample, the material was washed with mixture A, and the sample was

eluted in 30 µL of mixture B. The excess acetonitrile was evaporated in Concentrator Plus, and the final volume was adjusted to 12 µL (14 µL for proteome) with mixture A.

## MS measurement and data analysis

5 µL of sample were injected.

For BioID, the desalted and eluted peptides were loaded on an in-house packed C18 column (New Objective, 25 cm long, 75 µm inner diameter) for reverse-phase chromatography. The EASY-nLC 1200 system (Thermo Scientific) was mounted to a Q Exactive HF mass spectrometer (Thermo Scientific), and peptides were eluted from the column in an optimized 2 h gradient from 2–40% MS grade acetonitrile/0.5% formic acid solution at a flow rate of 225 nL min⁻¹. The mass spectrometer was used in a data-dependent acquisition mode with one MS full scan and up to 20 MS/MS scans using HCD fragmentation. MS scans were conducted with 60,000 resolution at a maximum injection time of 20 ms and MS/MS scans with 15,000 resolution at a maximum injection time of 75 ms.

For proteome analysis, the desalted and eluted peptides were loaded on an in-house packed C18 column (New Objective, 50 cm long, 75 µm inner diameter) for reverse-phase chromatography. The EASY-nLC 1200 system (Thermo Scientific) was mounted to an Orbitrap Exploris 480 mass spectrometer (Thermo Scientific), and peptides were eluted from the column in an optimized 90-min gradient of 2.4-32% acetonitrile/0.1% formic acid solution at a flow rate of 250 nl min⁻¹. The mass spectrometer was used in a data-dependent acquisition mode with one MS full scan followed by up to 20 MS/MS scans using HCD fragmentation. MS scans were conducted with 60,000 resolution at a maximum injection time of 28 ms and MS/MS scans with 15,000 resolution at a maximum injection time of 28 ms.

All raw files were processed with MaxQuant (for BioID ZNF524 WT vs ZF2 mutant: version 1.5.2.8; for proteome: version 1.6.5.0; for BioID ZNF524 WT vs NLS: version 2.0.1.0) using default settings unless specified otherwise and searched against the human UniProt database (95,934 entries, UP000005640, v20210307) or concatenated SwissProt/Trembl (proteome: 42,338 entries (SwissProt) + 54,436 entries (Trembl), UP000005640, v20200117). Carbamidomethylation (Cys) was set as a fixed modification, while oxidation (Met) and protein N-acetylation were considered variable modifications. For enzyme specificity, trypsin was selected with a maximum of two missed cleavages. Search results were filtered with a false discovery rate of 0.01, and for known contaminants, protein groups only identified by site, and reverse hits of the MaxQuant results. LFQ quantification (without fast LFQ) using at least 2 LFQ ratio counts (unique + razor peptides) and the match between run option were activated in the MaxQuant software.

## Immunofluorescence (IF) staining

Cells were seeded on coverslips to a maximum confluency of 70%. After overnight incubation, the cells were washed with DPBS and fixed to coverslips by 10 min incubation with 4% formaldehyde at RT. After washing with PBS (supplemented with 30 mM glycine for αTRF2/αGFP double staining), the cells were permeabilized with 0.5% Triton X-100, washed again and then blocked with 0.3% BSA (or 0.2% fish skin gelatin for αTRF2/αGFP double staining) in PBS (blocking buffer) for 1 h at RT. The respective primary antibody (see Supplementary Data 6) was diluted in blocking buffer and added to cells for 1 h incubation at RT or overnight at 4 °C. Subsequently, the cells were washed three times before addition of secondary antibody (see Supplementary Data 6) diluted in blocking buffer and 1 h incubation at RT. Following three washes with blocking buffer and one wash with PBS, the coverslip with the specimen was mounted onto the microscope slide using DAPI ProLong Diamond Antifade Reagent (Thermo Scientific) or Vectashield containing DAPI (Vector Laboratories). The slides were stored in the dark at RT, sealed and stored long-term at 4 °C.

For quantification of shelterin complex members in interphase WT and ZNF524 KO clones, pictures were taken with a Leica TCS SP5 confocal microscope (pinhole 60.05 μm, 2x zoom). Z-stacks were taken with a distance of 0.13 μm between focal planes. The laser and gain settings were adjusted to the sample with the lowest signal intensity. Fiji (ImageJ) was used for quantification of signal intensities and areas of the telomere foci. Therefore, the channels were split into the DAPI and red and green channels. A mask of the image was created to infer the volume of the imaged object. The threshold function of the software was used with activated plugins for the identification of round objects (Otsu). After setting the threshold for the image in the histogram settings, the z-stack was converted to a binary mask, and using the 3D OC Options menu, the integrated density was calculated. Additionally, the 3D Object counter menu was used, and the filters were set to a minimum of 4. An additional filter to remove the lowest 10% was applied for stringency.

For quantification of colocalization events, we used a Zeiss LSM 880 with a 100x/1.4 oil objective. Z-stacks were taken with 0.5 μm between focal planes. The images were analyzed with Fiji (ImageJ). After maximum intensity projection, the channels were split, and telomeric foci were counted. Subsequently, the number of GFP foci overlapping with telomeric foci was visually determined.

## Fluorescence in situ hybridization (FISH)

Cells were seeded 24 h prior to staining. After washing the slides with PBS, the cells were fixed with 4% formaldehyde for 10 min and washed with PBS. U2OS cells were incubated with permeabilization buffer (20 mM Tris-HCl, pH 8.0, 3 mM MgCl$_2$, 300 mM sucrose, 0.5% Triton X-100, 50 mM NaCl) at 37 °C for 1 h, and all other cell lines were treated with 0.5% Triton X-100 in PBS at RT for 7 min. The sample was dehydrated by successive immersion in 70%, 85% and 100% ethanol for 3 min each. For U2OS cells, the TAMRA-labeled C-rich telomere probe (#507207, Eurogentec) was diluted in hybridization buffer (3x SSC, 50% formamide, 10% (w/v) dextran sulfate, 50 μg mL$^{-1}$ heparin, 100 μg mL$^{-1}$ yeast tRNA, 100 μg mL$^{-1}$ salmon sperm DNA). For all other cell lines, the probe was diluted 1:100 in 1x blocking reagent by Roche and added to the slides, which were subsequently heated to 85 °C for 3 min. After either 4 h at RT or overnight incubation at 37 °C in a humidity chamber. In the case of U2OS, the cells were washed with wash buffer 1 (2x SSC, 50% formamide) at 37 °C for 20 min followed by wash buffer 2 (50 mM Tris-HCl pH 7.5, 150 mM NaCl, 0.05% TWEEN20) at RT for 15 min. All other cells were washed twice with wash buffer A (70% formamide, 10 mM Tris-HCl, pH 7.2), three times with wash buffer B (50 mM Tris-HCl, pH 7.5, 150 mM NaCl, 0.05% Tween 20) and twice with PBS. For TIF, cells were blocked (for U2OS: 10% FBS, 0.1% Triton X-100 in PBS; for HeLa: 0.3% BSA in PBS) for 1 h at RT and incubated with a 1:500 dilution of 53BP1 antibody (NB100-304, Novus) for 2 h at RT followed by PBS washes and incubation with a 1:300 diluted secondary antibody (goat anti-rabbit coupled to Alexa488, Thermo Scientific). After a final PBS wash, DAPI ProLong Diamond Antifade Reagent (Thermo Scientific) was added to the cells.

We analyzed the count data of TIF events using a generalized linear mixed model for negative binomially distributed data. For this purpose, we used the R package lme4. The factor genotype was implemented as a fixed effect. The factor clone was implemented as a random effect and played the role of a random perturbation of the fixed effect. The p-value for the influence of the factor genotype on the expected count was calculated using a likelihood ratio test. The confidence intervals were calculated by endpoint transformation from Wald-type confidence intervals for linear combinations of the (fixed) model parameters. They can be interpreted as confidence intervals for the expected number of TIFs in the WT/KO group, with the random effect of the factor clone removed[73].

## Chromosome orientation FISH

U2OS WT and KO clones were seeded in DMEM without antibiotics. After 8 h, BrdC (J65456.03, Alfa Aesar, Fisher Scientific) and BrdU (B5002-1G, Sigma Aldrich) were added at a 1:1,000 dilution and incubated at 37 °C for 10 h followed by treatment with 200 mM nocodazole for 8 h. Mitotic cells were harvested and exposed to hypotonic shock (10 mM sodium citrate, 25 mM KCl) at 37 °C for 30 min followed by fixation in cold methanol/acetic acid (3:1 v/v). Finally, the metaphases were spread on microscope slides, treated first with 0.5 mg mL$^{-1}$ RNase A at 37 °C for 10 min and then with 0.5 μg mL$^{-1}$ Hoechst 33258 (Sigma) in 2x SSC at RT for 15 min. The slides were exposed to 365 nm UV light, and the damaged BrdU/BrdC-substituted DNA strands were subsequently digested by 800 U Exonuclease III (Promega) in dedicated buffer (50 mM Tris-HCl pH 8.0, 5 mM MgCl$_2$, and 5 mM DTT) at 37 °C for 30 min. The metaphases were washed in PBS, dehydrated in a series of 70%, 85% and 100% ethanol and air-dried. The metaphases were hybridized first with Cy3-labeled G-rich telomere probe (1:100 dilution 5 nmol, PN-TG050-005, Eurogentec) and then with FITC-labeled C-rich telomere probe (1:100 dilution of 5 nmol, PN-TC011-005, Eurogentec) in the dark at RT for 1.5 h. The slides were washed (70% formamide, 10 mM Tris-HCl pH 7.4). Again, dehydration was performed in the previously mentioned ethanol series followed by drying. The metaphases were mounted with DAPI ProLong Gold Antifade Reagent (Thermo Scientific).

As previously described for the count data of TIF events, we analyzed the count data of t-SCE events in metaphase cells using a generalized linear mixed model for negative binomially distributed data.

## Protein purification for antibody generation and purification

His-MBP-ZNF524 was expressed from pCoofy4[74] for immunization, while His-ZNF524 was expressed from pCoofy1[74] for antibody purification from serum. The *E. coli* BL21 pRARE strain carrying the expression vector was grown in 8 × 1 L LB medium, and expression of the recombinant protein was induced with 0.5 M IPTG. After 3 days at 18 °C, the bacteria were pelleted at 5,000 g for 30 min, and the bacterial membrane was disrupted in 250 mL lysis buffer (50 mM Tris-HCl pH 7.5, 150 mM NaCl, 5% glycerol, 2 mM β-mercaptoethanol, 20 mM imidazole, 2 mM MgCl$_2$, 0.1 mM ZnCl$_2$, sm-nuclease and complete protease inhibitor cocktail by Roche) by sonication (Branson sonifier; duty cycle: 40; output control: 6; 2 × 3 min). The lysate was kept on ice at all times. To remove any cell debris, the lysate was centrifuged at 45,000 g for 30 min, and His-(MBP-) ZNF524 was further purified on a HisTrap™ HP (GE Healthcare) via the Akta Prime Plus System. After washing with 6% and 12% elution buffer (50 mM Tris-HCl pH 7.5, 150 mM NaCl, 5% glycerol, 2 mM β-mercaptoethanol, 300 mM imidazole), the His-tagged target protein was eluted with 100% elution buffer at a 1 mL min$^{-1}$ flow rate, and the collected fractions were checked for recombinant ZNF524 expression on 4-12% NuPAGE Novex Bis-Tris precast gels (Thermo Scientific). His-MBP-ZNF524 of ~75% purity was sent to Pineda Antikörper-Service, Berlin, for immunization of rabbits.

For antibody purification from rabbit serum, the elution fractions containing His-ZNF524 were further purified on a HiTrap Heparin HP™ (GE Healthcare). Therefore, the eluted protein fractions were diluted in 10 volumes of Buffer E (50 mM Tris-HCl pH 7.5, 2 mM DTT, 5% glycerol), applied to the column and eluted over a gradient of 200 mM NaCl up to 1 M NaCl. Fractions containing His-ZNF524 were dialyzed to coupling buffer (50 mM Tris-HCl pH 8.5, 5 mM EDTA) for storage at 4 °C. Antibodies against ZNF524 were purified and enriched from the serum against this recombinantly expressed His-ZNF524 using the SulfoLink® Immobilization Kit for Peptides (Thermo Scientific) according to the manufacturer's instructions.

## Generation of knockout cells

For the generation of ZNF524 KO clones, three guide RNAs targeting different regions in exon 2 of the ZNF524 gene were designed (Supplementary Data 5). DNA oligonucleotides of these regions were cloned and inserted into the PX459 V2 vector containing both the gRNA scaffolding and the Cas9 expression cassette. pSpCas9(BB)−2A-Puro (PX459) V2.0 was a gift from Feng Zhang (Addgene plasmid # 62988; http://n2t.net/addgene:62988; RRID:Addgene_62988)[75]. Successful cloning was confirmed by Sanger sequencing (GATC). 48 h after transfection, the cells were selected (3 µg mL⁻¹ puromycin), expanded and single-cell sorted on a BD FACSAria III SORP. Successful cleavage was confirmed on the unsorted pool by T7E1 assay. In short, gDNA was isolated using the QIAamp DNA Mini Blood Kit (Qiagen), and the target regions were PCR amplified using specific primers (Supplementary Data 5) followed by denaturation for 10 min at 95 °C and ramped reannealing (95-85 °C at −2 °C s⁻¹, 85-25 °C at −0.1 °C s⁻¹). Treatment with 10 units of T7 endonuclease I for 30 min at 37 °C revealed genomic modifications as visualized on a 2% agarose gel. Single cells were expanded and checked for ZNF524 expression using our self-produced α-ZNF524 antibody. The clonal lines that were negative for ZNF524 expression were subjected to next-generation sequencing to determine the genomic modifications. Genomic DNA of cell clones was extracted as described, and a 948 bp target region around the cut locus was amplified by PCR using the T7E1 assay primers, followed by a second PCR using P5 and P7 overhang primers (Supplementary Data 5). PCR products were purified using AMPure XP beads (Beckman Coulter), and the DNA concentration was determined using the Qubit dsDNA HS Assay Kit (Thermo Scientific). In a third PCR, index primers specific for each sample were used, and DNA was again purified with AMPure XP beads. All samples were pooled in an equimolar ratio and sequenced on a MiSeq Nano Flowcell, paired-end for 2×159 cycles plus 7 cycles for the index read. DNA-Seq measurements of U2OS WT and ZNF524 KO samples yielded on average 57 K reads of 159 nt length per sample. We assessed the quality of the sequenced reads with fastqc[76]. Adapter sequences were removed from both ends of both reads using cutadapt version 1.14[77]. Paired reads were merged using pandaseq version 2.11[78] with the following parameters: -d BFSrk -A pear. Merged reads were mapped to chromosome 19 of the *Homo sapiens* GRCh38 reference genome using gmap version 2017-02-15[79] with the following parameters: --min-intron length=200 -f sample --nofails. For localization and visualization of the mutations, we summarized mapped sequences using R version 3.4.3[80] and the CrispRVariants bioconductor package version 1.6.0[81]. Variants within the region of interest were localized, and mutation rates of all alleles were calculated for each sample.

## Southern blot analysis of telomere restriction fragment (TRF) lengths

Genomic DNA was isolated using the QIAamp DNA Blood Mini Kit (Qiagen) following the manufacturer's instructions. TRF length analysis was performed using the TeloTTAGG telomere length assay kit (Sigma-Aldrich) with slight modifications to the manufacturer's instructions[82]. 8 µg and 2.5 µg of DNA were digested for HeLa and U2OS cell line, respectively, using 20 U of HinfI and RsaI each at 37 °C for 4 h or overnight. Digested HeLa DNA was then resolved on a 0.8% agarose gel at 120 V for 4 h in 1x TAE buffer, and the gel was visualized using RedSafe nucleic acid stain (iNtRON). For U2OS cells, the digested DNA was resolved on 0.8% low-melt megabase agarose for 24 h with an initial switch time of 0.2 s and a final switch time of 12 s at 4 V/cm for 15 s and visualized using EtBr. The gel was incubated in 0.25 M HCl for 20 min for depurination and rinsed twice with distilled water followed by incubation in denaturation solution (0.5 M NaOH, 1.5 M NaCl) twice for 20 min. Subsequently, the gel was rinsed with distilled water twice before two washes with neutralizing solution (0.5 M Tris-HCl, 3 M NaCl, pH 7.5) for 20 min each. The digested DNA was then transferred to a positively charged nylon membrane (Hybond, N+, Amersham, UK) overnight by capillary osmosis in the presence of 20x SSC (3 M NaCl, 0.3 M sodium citrate tribasic dehydrate, pH 7) and fixed by UV crosslinking at 120 mJ using a Stratalinker® UV Crosslinker (Stratagene). The membrane was rinsed twice with 2x SSC and incubated with prewarmed DIG Easy Hyb Granules for 1 h at 42 °C before hybridization with DIG-labeled telomere probe (1 µL/5 mL of Hyb Granules) for 3 h at 42 °C. Subsequently, the membrane was washed twice with stringent buffer 1 (2x SSC, 0.1% SDS) at RT for 5 min each, twice with prewarmed stringent buffer 2 (0.2x SSC, 0.1% SDS) at 50 °C for 20 min each and rinsed with 1x wash buffer provided in the kit for 5 min. The membrane was blocked with 1x blocking solution for 30 min at RT, followed by incubation with anti-DIG-AP antibody (1:10,000) diluted in blocking solution for 30 min at RT and subsequently washed twice with 1x washing buffer, 15 min each at RT. Following incubation in 1x detection solution for 5 min at RT, the TRF smear was detected using the digoxigenin luminescent detection (CDP star) system and developed on X-ray films. The average telomere length was calculated by comparison to the 1 kb plus DNA ladder provided in the kit using WALTER (Web-based Analyzer of the Length of Telomeres) for analysis of U2OS or telotool for HeLa cells[83,84].

## C-circle assay

Genomic DNA was isolated from U2OS WT and ZNF524 KO clones using a QIAamp DNA Blood Mini Kit (Qiagen) with RNase treatment. Following quantification with the Qubit dsDNA HS Assay Kit (Thermo Scientific) according to the manufacturer's instructions, 300 ng of DNA was digested using 10 U each of Hinf I and Rsa I at 37 °C for 2 h. Digested DNA (7.5 ng and 15 ng) was amplified with 7.5 U φ29 polymerase (NEB) in 1X φ29 buffer (NEB) supplemented with 2 mM of dATP, dGTP and dTTP (Thermo Scientific) each and 0.1 mg mL⁻¹ BSA for 6 h at 30 °C, followed by heat inactivation at 70 °C for 20 min. Reactions lacking either φ29 polymerase or gDNA template served as negative controls.

## Immunofluorescence analysis of metaphase chromosome spreads

Cells were treated with Karyomax Colcemid (10 µg mL⁻¹, Thermo Scientific) at a 1:100 dilution for 2 h before harvest. Hypotonic buffer (75 mM KCl) was added to the cell suspension and incubated for 5 min at RT. Subsequently, the cell suspension was cyto-spun on microscope slides at 1,000 g for 5 min and treated with ice-cold KCM buffer (20 mM NaCl, 120 mM KCl, 10 mM Tris-HCl pH 7.2, 1 mM EDTA pH 8.0, 0.1% Triton X-100) for 1 min. Next, the metaphases were extracted with KCM buffer supplemented with 0.5% Triton X-100 for 5 min and then blocked with 1% (w/v) BSA in KCM buffer for 5 min. Incubation with primary and secondary antibodies took place in a humidified container at 37 °C for 1 h with washes with KCM buffer in between. The metaphases were fixed with 4% formaldehyde for 10 min at RT and finalized with DAPI ProLong Gold Antifade Reagent (Thermo Scientific). For quantification of TRF2/RAP1 abundance on metaphase chromosomes in U2OS WT and ZNF524 KO clones, pictures were taken with a Zeiss Axio Imager M2 with a 100x/1.4 oil objective. The images were analyzed with AxioVision software (Zeiss), and fluorescent intensities were estimated for each chromosome end by determining the area under the curve (auc) for a vertical line drawn across each signal with background subtraction.

## esiRNA synthesis and transfection

esiRNAs were produced as previously published[85]. Briefly, optimal regions for designing esiRNAs were chosen using the Deqor design algorithm[86] to fulfill two criteria: to obtain the most efficient silencing trigger in terms of silencing efficiency and to obtain the lowest chances to cross-silence other genes. The most favorable fragments were used to design gene-specific primers (Supplementary Data 5) using the

Primer3 algorithm (https://bioinfo.ut.ee/primer3-0.4.0/). Two esiRNAs for ZNF524 were designed and synthesized. PCR products for esiRNA production were confirmed by Sanger sequencing (1st BASE).

For esiRNA transfection, 35,000 U2OS cells were seeded in 2 ml medium in 6-well plates (Eppendorf) the evening before transfection. 10 µl Lipofectamine RNAiMAX (Thermo Scientific) were diluted in 240 µl OptiMEM (Thermo Scientific) and incubated for 5 min at RT. In a separate tube, 1.5 µg esiRNA were diluted in 250 µl OptiMEM. The solutions were combined, mixed and incubated for 20 min at RT, after which the transfection mix was mixed with the cell culture medium and added back to the corresponding well. 4 h post transfection, the cells were washed with 1x PBS and supplemented with 4 ml of fresh medium.

## qPCR
Total RNA was extracted from cells using the RNeasy Mini Kit (Qiagen) according to the manufacturer's instructions, including on-column DNA digestion with DNAseI. Subsequently, RNA was reverse transcribed into cDNA using the First strand cDNA synthesis Kit (Thermo Scientific) according to the manufacturer's instructions. For qPCR, we used Power SYBR™ Green PCR Master Mix (Thermo Fisher Scientific) with ZNF524-specific primers and GAPDH primers as internal control (Supplementary Data 5). Measurements were performed on an Applied Biosystems ViiA™ 7 Real-Time PCR System (Thermo Fisher Scientific).

## Next-generation RNA sequencing (RNA-seq)
Total RNA was extracted from cells using the RNeasy Mini Kit (Qiagen) according to the manufacturer's instructions, including on-column DNA digestion. NGS library prep was performed with Illumina's TruSeq stranded mRNA LT Sample Prep Kit following Illumina's standard protocol (Part # 15031047 Rev. E). Libraries were prepared by using only ¼ of the reagents with a starting amount of 250 ng, and they were amplified in 11 PCR cycles. Libraries were profiled in high-sensitivity DNA on a 2100 Bioanalyzer (Agilent Technologies) and quantified using the Qubit dsDNA HS Assay Kit in a Qubit 2.0 Fluorometer (Thermo Scientific). Libraries were pooled in equimolar ratio and sequenced on a 1 NextSeq 500 HighOutput Flowcell, SR for $1 \times 75$ cycles plus $2 \times 8$ cycles for dual index read.

mRNA read processing and mapping: Library quality was assessed with FastQC version 0.11.8 before being aligned against the *H. sapiens* genome assembly Homo_sapiens.GRCh38.98 and its associated .GTF and .BED files annotations. Such alignment was performed with STAR[87] version 2.7.3a (options: --runMode alignReads --outStd SAM --outSAMattributes Standard --outSJfilterReads Unique --outSAMunmapped Within --outReadsUnmapped None --outFilterMismatchNoverLmax 0.04 --outFilterMismatchNmax 999 --sjdbOverhang 75)[87]. Reads mapping to annotated features in the .GTF file were counted with featureCounts[88] version 1.6.2 (options: --donotsort -t exon)[88]. Coverage tracks were generated with deepTools version 3.1 (bamCoverage --binSize 1 --skipNonCoveredRegions --normalizeUsing CPM) and plotted using Gviz on an R framework[69,89,90]. Finally, the overall quality of the reads and the alignment was assessed with MultiQC version 1.7[91].

Differential expression analysis: Further filtering and exploratory analysis were performed in an R framework including ggplot2[92]. Pairwise differential expression comparisons were performed with DESeq2[93]. Gene expression in RPKM was used to filter out individuals with a replicate average lower than 0, thus considering them as nonexpressed. Differentially expressed genes (DEGs) were selected with an adjusted *p*-value (FDR) of <0.01, and a threshold of at least a 1 $\log_2$-fold-change difference between conditions was applied. Overlapping genes between conditions were assessed for significance with a hypergeometric distribution test (*p*-value < 0.01) as implemented in R base stats.

## Coimmunoprecipitation
Expression of ZNF524-GFP WT and NLS-GFP in U2OS cells was induced by 48 h treatment with 500 ng mL$^{-1}$ doxycycline. After 24 h, the cells were transfected for FLAG-TRF2 expression as previously described. Cells were harvested and lysed in radioimmunoprecipitation assay (RIPA) buffer (1% Igepal, 0.1% sodium deoxycholate, 150 mM NaCl, 50 mM Tris HCL pH 7.5) supplemented with cOmplete protease inhibitor by Roche. The GFP-tagged proteins were targeted using 10 µL GFP-Trap magnetic beads (Chromotek) per IP equilibrated in 10 mM Tris HCl, pH 7.5, while FLAG-TRF2 was targeted by αFLAG in PBB buffer. Per IP, 400 µg of lysate were diluted in the respective IP buffer and incubated on a rotating wheel at 4 °C overnight. Samples incubated with FLAG were subsequently supplemented with 12.5 µL of Dynabeads Protein G (Invitrogen) per IP and incubated for another 2 h. Using a magnetic rack, each sample was washed three times with the respective IP buffer and finally eluted in LDS buffer followed by 10 min at 70 °C.

## Reporting summary
Further information on research design is available in the Nature Portfolio Reporting Summary linked to this article.

## Data availability
All data needed to evaluate the conclusions in the paper are present in the paper and/or the Supplementary Materials. The atomic coordinates and structure factors for ZNF524 ZF1-4 in complex telomeric DNA have been deposited in the Protein Data Bank (PDB) under the accession code PDB 7YSF. The mass spectrometry data have been deposited to the ProteomeXchange Consortium via the PRIDE[94] partner repository with the dataset accession codes PXD024630, PXD043084 and PXD031416. The ChIP-seq data are available under the GEO accession code GSE196661. The RNA-seq data are available in the SRA, BioProject under the accession code PRJNA804430. Source data are provided with this paper. All unique biological materials are readily available from the authors upon request. Source data are provided with this paper.

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

## Acknowledgements

Assistance by the following IMB Core facilities is gratefully acknowledged: the Microscopy Core Facility, the Genomics Core Facility, the Protein Production Core Facility, the Flow Cytometry Core Facility, and the Media Lab. We thank Fridolin Kielisch from the Bioinformatics Core Facility for invaluable advice on the statistical analysis. We thank the Microscopy Core at CSI Singapore. We also thank the staff of the BL17U, BL18U1 and BL19U1 beamlines at the National Center for Protein Science Shanghai and Shanghai Synchrotron Radiation Facility for assistance during data collection. pLIX_403 was a gift from David Root, and pInducer20 was a gift from Stephen Elledge. This study was funded by the following: Deutsche Krebshilfe [70112401] (to F.B.); Wilhelm-Sander-Stiftung [2020.074.1] (to F.B.). Singapore Agency for Science, Technology and Research (A*STAR) (to O.D.); National Research Foundation Singapore and the Singapore Ministry of Education under its Research Centres of Excellence initiative (to D.K.); NUHS Seed Fund [NUHSRO/2020/105/T1/Seed-Mar/01] (to D.K.). NMRC Open Fund Individual Research Grant [MOH-OFIRG21jun-011] (to D.K.); Strategic Priority Research Program of the Chinese Academy of Sciences [XDB39010300]

(to Y.S.). Chinese National Natural Science Foundation [92149302, 32090040, 31870760] (YS); Ministry of Science and Technology of China [2019YFA0508403, 2016YFA0500700] (to Y.S.). Chinese National Natural Science Foundation [U1932122, 32171222, 32371282] (to F.L.); Chinese National Natural Science Foundation [32200979] (to Y.Y.).

## Author contributions

F.B. and D.K. initiated the research. F.B., D.K. and H.B. conceptualized and planned the research. H.B. performed and analyzed most experiments with help from F.C., N.V., G.R., L.L., A.H., F.R., A.F.S., Z.W.S., W.K.Y. and V.K. Z.X. and F.L. performed and analyzed the ITC and crystallography experiments with help from Y.Y. and Y.S. M.L. performed data analysis and helped with visualization. O.D. provided expertise for the BioID assay. F.B., D.K. and H.B. wrote the manuscript with input from all authors.

## Competing interests

The authors declare no competing interests.
