## [Peer Review File · Nature Communications]

ZNF524 directly interacts with telomeric DNA and supports telomere integrityREVIEWER COMMENTS

Reviewer #1 (Remarks to the Author):

This work focuses on the new factor ZNF524 and show the specific motif by which it binds to telomeric repeats through crystal structure resolution. The authors further show that it directly binds to NNNGGG repeats, co-localizes with the telomeres and mediates TRF2/RAP1 subcomplex at telomeres. Loss of ZNF524 results in an increase in DNA damage and telomere recombination. This manuscript is of importance, however, there are several concerns and questions before this manuscript is accepted:

Major comments:

1. Fig. 1: Throughout this figure, the data seems to support that ZNF547 has a specificity to NNNGGG sequence. A sequence that includes repeats of GGG can form G-quadruplexes. Can ZNF524 bind G-quadruplexes?
2. Figure 1 b-d since all of these gel pictures were taken using X-ray film, the contrast used to achieve a white background is too extreme. Please provide less or unmanipulated pictures of the scans to appropriately assess these figures for background.
3. Fig.1 d: This subfigure had multiple issues that should be resolved:
 - a. Adding a tag to generate a recombinant protein may cause changes to its conformation and thereby its activity, localization, or function. The recombinant His-ZNF546 seem less specific to the NNNGGG sequence and bind to a non-telomeric random sequence as strongly as the positive control (10% input). This could be due to the specificity of the His antibody, but otherwise, it is very important to confirm that the tag did not change the WT specificity in order to make conclusions from the followed experiments performed with it. The Flag tag used in Figure 1b looks much more convincing and can be clones into a bacterial plasmid. Repeating this with a Flag or a GST tag will probably give cleaner data.
 - b. The telomeres are double stranded and have a 3' single strand tail. Do you think ZNF547 binds to the single strand portion of the telomeres? This would be interesting as not many factors bind the single strand of the telomeres other than POT1. If not, double strand probe should be tested.
 - c. Additionally, it is not clear what is the second lower band that seem to be partially cropped around 37 KDa? It seem darker only when TTAGGG telomere sequence is used.
4. Page 5 (lines 138-14) + Figure 3: "In agreement with the in vitro data, the DNA-binding deficient ZNF524-GFP ZF2 mutant (C144A) did not form foci and distributed evenly in the nucleus (Fig. 3a)." The data presented in Figure 3a does not demonstrate this statement and the mutant ZNF524-GFP is not "evenly distributed in the nucleus". The bright cells stained with ZNF527 may be cells right after M phase and this heavy staining may suggest that it is replication stage specific. For this paper, suggest revising to: "the DNA-binding deficient ZNF524-GFP ZF2 mutant (C144A) showed smaller and more diffused foci with loss of clear colocalization with the telomeres (Fig. 3a)."
5. Supplementary Figure 4e and g. These southern blot results are confusing. The clonal effect on the telomere length in both WT and KO cells is very significant. Any factor important for telomere integrity or for shelterin function may cause gradual and cumulative effect on the telomeres with increased cell replication during cloning and cell passaging. A total population of un-cloned knockout cells (in replicates) may give a more accurate data.
6. Figure 5c-d: Based on this figure, there is asymmetry between the FITC-labelled C-rich telomere probe (green) and the red G-rich probe. Asymmetry between the C-rich and the G-rich telomeres may be the result of asymmetric telomere elongation of processing. This may be an important finding C-rich probe marks e 3' G-rich telomere overhang and therefore, should be quantified for both, the red and the green probes in WT and KO cells.

Minor comments:

Small typos should be corrected throughout the manuscript Supplementary Fig. 7 is actually Supplementary Fig. 4, "cclone" in the legend of Supplementary Fig. 7

Reviewer #2 (Remarks to the Author):

In this manuscript Braun et al., follow up on their previous work that identified ZNF524 as a member of the telomere-binding proteome. Here, they use a combination of DNA pull-down assays and isothermal calorimetry to validate the interaction between ZNF524 with telomeres and determine the relative contributions of the individual zinc finger domains. Overall, they find that ZF2 is particularly important for the interaction with telomeres. Next, they solve the crystal structure of ZNF524 zinc fingers bound to telomeric DNA. Further supporting the ZNF524-telomere interaction, they show that ZNF524 colocalizes with telomeres in ALT cells. Additionally, using BioID proteomics they show that ZNF524 is in close proximity to known telomere binding proteins including TRF1 and TRF2. The remainder of the data focus on a proposed function of ZNF524 in the regulation of shelterin factors binding to telomeres and the activation of telomere damage responses and homologous recombination. To this end, the authors show that both RAP1 and TRF2 have reduced focal intensity following either short-term depletion or in ZNF524 KO clones. Further, they show that WT ZNF524 can rescue the reduction of TRF2 focal intensity. Additionally, they observe mild increases in both telomere damage responses and telomere recombination (t-SCEs).

This paper appears to be methodologically sound and the first few figures examining ZNF524 binding to telomere DNA and the ZNF524-telomere structure are impressive. However, there are a variety of issues with the later figures in the paper as follows:

- The observation that TRF2/RAP1 binding to telomeres is reduced upon ZNF524 loss, which is a major conclusion of the paper, is questionable for the following reasons:
 - 1) It is claimed in the paper that the changes in TRF2/RAP1 foci intensity are not due to TRF2 protein levels or telomere length. However, when the authors examined TRF2 or RAP1 protein levels by western blotting they find that they are in fact reduced. These findings do not quite reach statistical significance, but the magnitude of the effect is very similar to the observed reduction in TRF2 foci brightness. Further, the decrease in TRF2/RAP1 is also observed in the authors proteomic data (Supp figure 6C). Similarly, when telomere length is evaluated in ZNF524 KO clones across two different cell lines and using two different assays it is found to be consistently reduced (Supp figure 4: In some cases the observations reach statistical significance and in other cases the p-value is just above 0.05). As both a decrease in TRF2/RAP1 protein and a decrease in telomere length would be expected to result in lower foci intensity of TRF2/RAP1 without necessarily altering the ability of TRF2/RAP1 to bind to telomeres these results raise significant questions as to the claim of reduced TRF2/RAP1 binding to telomeres. Is the reduction simply the result of decreased telomere length?
 - 2) The statistical analysis of TRF2/RAP1 foci seems to be unorthodox - it appears that individual foci were considered as replicates, which is inappropriate.
 - 3) Even if the changes in TRF2/RAP1 foci intensity can be trusted, the magnitude of these effects is quite small and it is unclear that this would lead to consequential changes in telomere biology.
 - 4) Data suggesting why/how TRF2 is being affected is lacking. This would help to convince that TRF2 binding to telomeres is actually reduced.

This veracity of this claim is very important to the paper and therefore needs to be confirmed. This can be done relatively easily by evaluating the ratio of TRF2/RAP1 to the telomere probe under conditions of ZNF524 loss. Additionally, it should be made clear how the statistics were done, and it would be helpful if a more conventional presentation of the TRF2 foci intensity were used (considering

each clone as a biological rep in a simple bar graph).

- The observations of increased telomere damage and telomere homologous recombination (cross-over events measured via t-SCEs) are very modest (mean changes of ~ 0.5 -1 event/cell). These types of subtle changes could easily be a secondary effect of ZNF524 loss (for instance any change in cell cycle dynamics, global DNA damage repair, replication stress, or transcriptional effects of ZNF524, could alter the actual or apparent number of telomere damage responses subtly). It seems that these marginal effects are unlikely to be primarily attributed to ZNF524. The lack of change in C-circles in this context should also be discussed.

- A mechanistic explanation for the results/functions of ZNF524 are not given. While the biochemical/structural biology aspects of this paper are well done, it is of only minimal interest to the field without a more thorough explanation of the functional role of ZNF524 at telomeres. Such a mechanistic explanation could involve the proposed loss of TRF2 at telomeres, or perhaps telomere shortening and/or reduced shelterin expression, however this would require additional exploration. It is of note that the authors generate several datasets that could be used as starting points for determining the mechanistic role of ZNF524 (ZNF524 BioID, ZNF524 KO proteome), and there is some discussion of the fact that ZNF524 interacts strongly with a protein involved in chromatin remodeling (DPY30), however these leads are not further explored.

Minor points:

- Include a western blot confirming esiRNA knockdown.
- Line 220 refers to ZNF524 depletion, but the related figures are of knockout.

Reviewer #3 (Remarks to the Author):

The manuscript by Braun et al described a zinc finger protein ZNF524 that locates on telomeric DNA to maintain telomere integrity. The authors have performed biochemical and structural analyses to demonstrate that ZNF524 directly binds telomeric repeats with high affinity in vitro and in vivo through its four zinc fingers for base-specific recognition. They also provide genetic evidence showing that ZNF524 maintains the TRF2/RAP1 subcomplex at telomeres thus preventing genomic instability. The biochemical and structural data are informative. However, some of the results obtained in the functional studies are not convincing, and may require more validations.

1. The authors claimed that ZNF524 directly binds to telomeric TTAGGG repeats with nanomolar affinity in vitro. Surprisingly, ZNF524 localizes to telomeres only in ALT cells but not in telomerase cells such as HeLa 1.3 that harbors long telomeres comparable to that in U2OS. What is the underlying mechanism that prevents ZNF524 from binding to telomeres in telomerase positive cells.
2. Fig 3a and b. The authors pointed out that in >90% of U2OS cells co-localization of ZNF524 and TRF2 is visible with an average of 66% of all TRF2 foci being positive for ZNF524. A short statement of what TRF2 foci lack ZNF524 staining is desirable. Do they represent shorter telomeres? If not, can the authors think of anything that might have caused it?
3. Fig 3c and d. Given that ZNF524-GFP directly associates with more than half telomeres in a cell with high affinity, the ~ 2.5 x enrichment of telomere DNA signal over background revealed by the ChIP assay implies that the IP efficiency is poor. Could it be due to the overexpression of ZNF524-GFP or the use of too few antibodies in the experiment? Would it be possible to perform the experiment using antibodies against endogenous ZNF524? In addition, ChIP with anti-TRF1/TRF2 is desired to serve as a positive control to better evaluate the enrichment of ZNF524.
4. Fig 4a and b. The immunofluorescence staining of TRF1 in U2OS cells is of poor quality and seems not suitable for further quantitative analysis.
5. Fig 4b. The change in the density of TRF2 signal between WT group and KO group seems mild when compared to the intra-group variance caused by the high clonal heterogeneity. This makes it less convincing that depletion of ZNF524 leads to reduction in TRF2/subcomplex at telomere. Moreover, it

seems that ZNF524 removal also causes a slight decrease in telomere length (Fig. S4f), which may be a reason for the mild change of TRF2 signal in the KO group.

6. Fig. S4e-h. Because of the high clonal heterogeneity, it seems inappropriate to directly compare the telomere length between WT and KO. Instead, comparison of telomere length among KO, KO+WT and KO+ZF2mut might be more informative.

7. The authors indicated that the function of ZNF524 is not restricted to ALT telomeres based on the observation that depletion of ZNF524 causes an increase in TIFs in both ALT (U2OS) and telomerase-positive (HeLa) cells. This observation is confusing since ZNF524 is not found at the telomere in HeLa cells. Will ZNF524 removal leads to decreased telomere-bound TRF2 and Rap1 in HeLa cells? What are the actual functions of ZNF524 in cells? Are the functions dependent on the telomere binding?

REVIEWER COMMENTS (BLACK)

AUTHOR RESPONSE (BLUE)

We would like to thank all reviewers for the constructive feedback and the recognition of both the importance of our findings and the quality of our work.

Reviewer #1 (Remarks to the Author):

This work focuses on the new factor ZNF524 and show the specific motif by which it binds to telomeric repeats through crystal structure resolution. The authors further show that it directly binds to NNNGGG repeats, co-localizes with the telomeres and mediates TRF2/RAP1 subcomplex at telomeres. Loss of ZNF524 results in an increase in DNA damage and telomere recombination. This manuscript is of importance, however, there are several concerns and questions before this manuscript is accepted:

Major comments:

1. Fig. 1: Throughout this figure, the data seems to support that ZNF547 has a specificity to NNNGGG sequence. A sequence that includes repeats of GGG can form G-quadruplexes. Can ZNF524 bind G-quadruplexes?

This might be a misunderstanding. While ZNF524 also recognises telomeric variant repeat sequences, the quantified affinities (Supplementary Fig. 1c) are >3-fold lower compared to the 90 nM affinity determined for binding to TTAGGG. This is reminiscent of ZBTB48 (also known as TZAP) for which our labs had also documented an association with TCAGGG, TGAGGG and TTGGGG repeats in vitro (with less enrichment compared to TTAGGG; Fig. 1C in Jahn et al., 2017, PMID: 28500257) and subsequently determined quantitatively similar differences in K_D values (Fig. 1K/L in Zhao et al., 2018, PMID: 29134956).

However, to test a possible interaction with telomeric G quadruplexes, we annealed a single-stranded telomeric oligo in the presence of potassium and verified G quadruplex formation by CD spectroscopy (**new. Suppl. Fig. 1e**). Subsequent ITC measurements quantified a K_D value of 3.58 μ M (**new Suppl. Fig. 1d, updated Suppl. Table 2**), which is ~40-fold higher than the affinity to double-stranded telomeric DNA. This suggests that ZNF524 does not bind G-quadruplexes in agreement with previous mass spectrometry screens aimed at identifying proteins binding to telomeric G-quadruplexes (e.g. Williams et al., 2017, PMID: 28859475)

This result further aligns with our co-crystal structure in Fig. 2, illustrating that the base-specific contacts 1) cover a larger binding sequence, namely GNNAGGGTT, and 2) include base-specific interactions on both DNA strands. In combination with the alignment of the zinc fingers along the major groove, we are here unambiguously reporting a direct interaction with telomeric dsDNA.

2. Figure 1 b-d since all of these gel pictures were taken using X-ray film, the contrast used to achieve a white background is too extreme. Please provide less or unmanipulated pictures of the scans to appropriately assess these figures for background.

The reviewer might have missed it, but as indicated in the methods section (Western blots) we did not use X-ray films for the Western blot data in Fig. 1b-d, but these blots were imaged using a ChemiDoc Imaging System (Biorad). The appearance of digital blots is distinct from the background in X-ray films and we are already reporting unmanipulated pictures.

3. Fig.1 d: This subfigure had multiple issues that should be resolved:

a. Adding a tag to generate a recombinant protein may cause changes to its conformation and thereby its activity, localization, or function. The recombinant His-ZNF546 seem less specific to the NNNGGG sequence and bind to a non-telomeric random sequence as strongly as the positive control (10% input). This could be due to the specificity of the His antibody, but otherwise, it is very important to confirm that the tag did not change the WT specificity in order to make conclusions from the followed experiments performed with it. The Flag tag used in Figure 1b looks much more convincing and can be clones into a bacterial plasmid. Repeating this with a Flag or a GST tag will probably give cleaner data.

The 10% input lane is not a positive control but rather serves as a loading control for cases where no enrichment is detectable in any of the experimental lanes (e.g. see FLAG-ZNF524 ZF2 mut C144A in Fig. 1c). While there is some background signal visible in the GTGAGT sample in this panel, this is common in biochemical assays at the intersection of stringency (e.g. salt conditions) as well as protein and bait quantities. Importantly, there is obvious enrichment on TTAGGG over the GTGAGT negative control with weaker enrichment on the TCAGGG, TGAGGG and TTGGGG variant repeat sequences.

These findings are further corroborated in Supplementary Fig. 1c with K_D value quantifications. For clarity we have therefore moved the original Fig. 1d to Supplementary Fig. 1 (**now Suppl. Fig. 1b**) to have the Western blot and the K_D value quantifications side-by-side. This should also increase the emphasis in the main Fig. 1 on the actual quantitative data

with purified ZNF524 in the ITC measurements and the contribution of individual zinc fingers (original Fig. 1e, **now Fig. 1d**).

b. The telomeres are double stranded and have a 3' single strand tail. Do you think ZNF547 binds to the single strand portion of the telomeres? This would be interesting as not many factors bind the single strand of the telomeres other than POT1. If not, double strand probe should be tested.

Similar to the G-quadruplexes above, we have now included ITC measurements for single-stranded telomeric DNA. Here, we could not detect any quantifiable K_D value (**new Suppl. Fig. 1d-e, updated Suppl. Table 2**), suggesting that ZNF524 is not able to bind to the single-strand portion of telomeres.

The data presented in Fig. 1 and Fig.2 was already based on double-stranded probes. In the pull-down assays (Fig. 1b-d) we included a 2nt overhang that allows for concatemerization. The probes used for both the ITC measurements (original Fig. 1e, **now Fig. 1d**) and the crystal structure (Fig. 2) contain a 1nt overhang, but as seen in the co-crystal structure in Fig. 2, ZNF524 interacts exclusively with the double-stranded portion and our ITC measurements suggest binding to telomeric dsDNA with high affinity at a K_D value of 90 nM.

c. Additionally, it is not clear what is the second lower band that seem to be partially cropped around 37 kDa? It seem darker only when TTAGGG telomere sequence is used.

The lower band around 37kDa is likely a minor degradation product, something that is not uncommon with recombinantly expressed/purified proteins. This band appears darker in the TTAGGG sample simply due to the more abundant enrichment compared to the other probes, but is visible in the TCAGGG, TGAGGG and TTGGGG samples as well proportionate to their respective enrichments.

4. Page 5 (lines 138-14) + Figure 3: “In agreement with the in vitro data, the DNA-binding deficient ZNF524-GFP ZF2 mutant (C144A) did not form foci and distributed evenly in the nucleus (Fig. 3a).” The data presented in Figure 3a does not demonstrate this statement and the mutant ZNF524-GFP is not “evenly distributed in the nucleus”. The bright cells stained with ZNF527 may be cells right after M phase and this heavy staining may suggest that it is replication stage specific. For this paper, suggest revising to: “the DNA-binding deficient ZNF524-GFP ZF2 mutant (C144A) showed smaller and more diffused foci with loss of clear colocalization with the telomeres (Fig. 3a).”

We understand the concern, but the suggested phrase might be somewhat misleading as well. What we are aiming to highlight here is that ZNF524-GFP ZF2 mut does not form discrete foci and rather has a diffuse distribution pattern in the nucleus. While this might not cover 100% of the nuclear surface area homogenously, the rephrasing suggested by the reviewer would imply the presence of some foci akin to telomeres. Here, we simply had nothing that would allow to quantify any putative co-localization with telomeres/TRF2.

We have therefore rephrased our original sentence to “...did not form foci and distributed **diffusely** in the nucleus (Fig. 3a)” (**page 5 line 148-149**). We hope that this wording captures the spirit of the reviewer’s recommendation.

5. Supplementary Figure 4e and g. These southern blot results are confusing. The clonal effect on the telomere length in both WT and KO cells is very significant. Any factor important for telomere integrity or for shelterin function may cause gradual and cumulative effect on the telomeres with increased cell replication during cloning and cell passaging. A total population of un-cloned knockout cells (in replicates) may give a more accurate data.

We appreciate the suggestion, but we respectfully disagree with this assessment. We here made the substantial effort of establishing multiple individual WT and ZNF524 KO clones to combine genetically clean samples with rigorous statistical evaluation. Pooled KOs are suboptimal because you invariably obtain a mixed population of KO and WT cells (or small in-frame deletions that are essentially WT) and in comparison to our single-cell sorted KO clones the suggested replicates would rather be technical in nature. We agree that there is heterogeneity in terms of telomere length across clones (more pronounced in HeLa due to the shorter average telomere length), but we would argue that it is important to document this heterogeneity when working with KOs and to include this biological variation as part of a rigorous statistical assessment. We would like to refer to a similar analysis in our previous work on ZBTB48 (Fig. 3A/B in Jahn et al., 2017, PMID: 28500257). Here, despite clonal heterogeneity a ZBTB48-dependent effect on telomere length could be established. This approach is in line with other publications in the field, e.g. Dille et al., 2016, PMID: 27760120.

6. Figure 5c-d: Based on this figure, there is asymmetry between the FITC-labelled C-rich telomere probe (green) and the red G-rich probe. Asymmetry between the C-rich and the G-rich telomeres may be the result of asymmetric telomere elongation of processing. This may be an important finding C-rich probe marks e 3' G-rich telomere overhang and therefore, should be quantified for both, the red and the green probes in WT and KO cells.

Across our images we did not notice a systematic difference for the C-rich and G-rich probes between the WT and KO cells. However, since our t-SCE measurements were not intended to quantify fluorescence signal intensity (i.e. qFISH) our images are not suitable for the proposed analysis. Importantly, some differences between the C-rich and G-rich probes are to be expected within the same sample/staining due to differences in secondary structures, hybridization efficiency etc.

Minor comments:

Small typos should be corrected throughout the manuscript Supplementary Fig. 7 is actually Supplementary Fig. 4, “clone” in the legend of Supplementary Fig. 7

Thanks for spotting this. We have corrected the numbering and “clone” typo in the corresponding figure legends.

Reviewer #2 (Remarks to the Author):

In this manuscript Braun et al., follow up on their previous work that identified ZNF524 as a member of the telomere-binding proteome. Here, they use a combination of DNA pull-down assays and isothermal calorimetry to validate the interaction between ZNF524 with telomeres and determine the relative contributions of the individual zinc finger domains. Overall, they find that ZF2 is particularly important for the interaction with telomeres. Next, they solve the crystal structure of ZNF524 zinc fingers bound to telomeric DNA. Further supporting the ZNF524-telomere interaction, they show that ZNF524 colocalizes with telomeres in ALT cells. Additionally, using BioID proteomics they show that ZNF524 is in close proximity to known telomere binding proteins including TRF1 and TRF2.

The remainder of the data focus on a proposed function of ZNF524 in the regulation of shelterin factors binding to telomeres and the activation of telomere damage responses and homologous recombination. To this end, the authors show that both RAP1 and TRF2 have reduced focal intensity following either short-term depletion or in ZNF524 KO clones. Further, they show that WT ZNF524 can rescue the reduction of TRF2 focal intensity. Additionally, they observe mild increases in both telomere damage responses and telomere recombination (t-SCEs).

This paper appears to be methodologically sound and the first few figures examining ZNF524 binding to telomere DNA and the ZNF524-telomere structure are impressive. However, there are a variety of issues with the later figures in the paper as follows:

- The observation that TRF2/RAP1 binding to telomeres is reduced upon ZNF524 loss, which is a major conclusion of the paper, is questionable for the following reasons:

- 1) It is claimed in the paper that the changes in TRF2/RAP1 foci intensity are not due to TRF2 protein levels or telomere length. However, when the authors examined TRF2 or RAP1 protein levels by western blotting they find that they are in fact reduced. These findings do not quite reach statistical significance, but the magnitude of the effect is very similar to the observed reduction in TRF2 foci brightness. Further, the decrease in TRF2/RAP1 is also observed in the authors proteomic data (Supp figure 6C).

We respectfully disagree with this assessment. The protein levels of TRF2 and RAP1 (as of the remaining shelterin proteins in the proteome data) do not reach significance and even when considering the existing variation, this does not reflect the reduction in TRF2/RAP1 association with telomeres that we report. We do acknowledge that smaller effects might be possible and those are not mutually exclusive with a dissociation from telomeres. In fact, many proteins lose stability when they lose complex membership or binding to their substrate or, as in this case, DNA.

However, we want to stress that we could not identify any statistically reproducible protein level changes. We assume that if we had made such a claim this reviewer would have rightfully pointed out our lack of evidence – this argument should not be used both ways.

Similarly, when telomere length is evaluated in ZNF524 KO clones across two different cell lines and using two different assays it is found to be consistently reduced (Supp figure 4: In some cases the observations reach statistical significance and in other cases the p-value is just above 0.05). As both a decrease in TRF2/RAP1 protein and a decrease in telomere length would be expected to result in lower foci intensity of TRF2/RAP1 without necessarily altering the ability of TRF2/RAP1 to bind to telomeres these results raise significant questions as to the claim of reduced TRF2/RAP1 binding to telomeres. Is the reduction simply the result of decreased telomere length?

We do not want to claim a change in telomere length given the lack of reaching generally accepted standards of statistical significance. However, even if we do consider the premise of shorter telomere length in the absence of ZNF524, in U2OS cells such a change would be in the ballpark of 1-2kb (Supplementary Fig. 5e). Relative to an average telomere length of >20kb this does not explain the reduction in TRF2/RAP1 association with telomeres. Furthermore, we could not detect changes in POT1 and TRF1 foci intensity (Fig. 4a/b & Supplementary Fig. 6a/b). Therefore, our data does not support the idea that the reduction is simply the result of decreased telomere length.

2) The statistical analysis of TRF2/RAP1 foci seems to be unorthodox - it appears that individual foci were considered as replicates, which is inappropriate.

We completely agree with the reviewer that it would have been inappropriate to use individual foci as replicates for our statistical analysis. While the violin plots are meant to transparently display the detailed data distribution, the statistics have indeed been exclusively done on the average values of the 5 WT and KO clones. Thanks to this comment, we realised that the figure legend might have been misleading. Therefore, we have added the n=5 information in the figure legends accordingly to avoid confusion with the amount of foci analysed.

3) Even if the changes in TRF2/RAP1 foci intensity can be trusted, the magnitude of these effects is quite small and it is unclear that this would lead to consequential changes in telomere biology.

While we see the reviewer's point, we respectfully disagree. The average changes of TRF2/RAP1 foci intensity across interphase and metaphase cells are up to 2-fold (please note the \log_2 scale for all IF quantifications). This corresponds to loss-of-heterozygosity for TRF2. As discussed in the manuscript these findings are in line with previously reported partial deprotection of telomeres/partial knock-down of TRF2 (Cesare et al., 2013, PMID: 23850488) and a quantitatively similar result was demonstrated upon Lamin B1 overexpression (Pennarun et al., 2021, PMID: 24469544). Therefore, our data demonstrates consequential changes in telomere biology in agreement with similar observations of other research groups.

4) Data suggesting why/how TRF2 is being affected is lacking. This would help to convince that TRF2 binding to telomeres is actually reduced.

We agree with the reviewer that it will be important to establish the mechanism behind the ZNF524-dependent reduction in TRF2/RAP1 subcomplex association with telomeres. However, we here already describe an entirely novel direct telomere-binding protein – an important finding in itself – and provide early evidence for functional impacts on telomere homeostasis.

To start addressing a putative mechanism, we have now identified an interaction between ZNF524 and the ChAHP complex and documented ZNF524-dependent changes in CBX3/HP1gamma abundance at telomeres (**new Suppl. Fig. 3a-c**). Please see below for further details on this point.

This veracity of this claim is very important to the paper and therefore needs to be confirmed. This can be done relatively easily by evaluating the ratio of TRF2/RAP1 to the telomere probe under conditions of ZNF524 loss. Additionally, it should be made clear how the statistics were done, and it would be helpful if a more conventional presentation of the TRF2 foci intensity were used (considering each clone as a biological rep in a simple bar graph).

Please see above for our detailed answer on these various points. Again, we represent the IF data with violin plots to transparently document the intercellular and intracellular variation in TRF2 foci signal intensity. Importantly, the averages (equivalent to what would be visible in a simple bar graph) are already part of these figures.

To further address this point, we have generated an independent dataset in which we have quantified the foci intensity of TRF1, TRF2 and RAP1 in the five independent U2OS WT and ZNF524 KO clones. As in the original manuscript, we again see that there is a significant reduction in TRF2 and RAP1 signal intensity at telomeres while TRF1 abundance is unaffected across the different clones. The data in Fig. 4 contains more detailed information including matched complementation with the expression of either ZNF524 WT or ZF mutant constructs in all give ZNF524 KO clones. If the reviewers and editor do not deem this redundant, we can also offer to display this independent dataset as a supplementary figure.

Figure to the reviewers 1: (a) Representative immunofluorescence images of U2OS WT and ZNF524 KO cells stained for TRF2 (green) and TRF1 (red). Scale bars represent 10 μ m. Nuclei were counterstained with DAPI (blue). (b) Quantification of TRF1 and TRF2 IF signals in WT and KO clones. The violin plot shows the individual data points as densities. The mean is indicated by a solid line; *p*-values are determined by Student's *t*-test and indicated above the plot; ** *p* < 0.01; *n* = 5 for all conditions. (c) Representative IF pictures of U2OS WT and ZNF524 KO cells stained for RAP1 (red). (d) Quantification of the RAP1 IF signal in WT and KO clones depicted as violin plots. The mean is indicated by a solid line; *p*-values are determined by Student's *t*-test (*n* = 5) and indicated above the plot; * *p* < 0.05.

- The observations of increased telomere damage and telomere homologous recombination (cross-over events measured via t-SCEs) are very modest (mean changes of ~0.5-1 event/cell). These types of subtle changes could easily be a secondary effect of ZNF524 loss (for instance any change in cell cycle dynamics, global DNA damage repair, replication stress, or transcriptional effects of ZNF524, could alter the actual or apparent number of telomere damage responses subtly). It seems that these marginal effects are unlikely to be primarily attributed to ZNF524. The lack of change in C-circles in this context should also be discussed.

The frequency of both TIFs and tSCEs are roughly doubled in U2OS ZNF524 KO clones. Given that both features are hallmarks of ALT cell lines, doubling their frequency does not appear modest/marginal to us.

The rather long list of secondary effects to be tested is beyond the scope of this first description of a new telomeric protein. Please note that we have nevertheless already made substantial efforts in this direction, e.g. transcriptional effects of ZNF524 have been tested. However, the absence of unique binding sites in the genome beyond telomeres (tested as part of the ChIP-seq data in Fig. 3e) and no significant change in even a single transcript in our RNA-seq data comparing U2OS ZNF524 WT and KO cells (Suppl. Fig. 7a) indicates that transcriptional regulation is not a contributor to the observed telomeric phenotypes. In addition, we have now measured cell cycle distributions across our 5 WT and KO clones each for both HeLa and U2OS and did not detect any significant differences, ruling out another putative secondary effect (**new Suppl. Fig. 5i**).

- A mechanistic explanation for the results/functions of ZNF524 are not given. While the biochemical/structural biology aspects of this paper are well done, it is of only minimal interest to the field without a more thorough explanation of the

functional role of ZNF524 at telomeres. Such a mechanistic explanation could involve the proposed loss of TRF2 at telomeres, or perhaps telomere shortening and/or reduced shelterin expression, however this would require additional exploration. It is of note that the authors generate several datasets that could be used as starting points for determining the mechanistic role of ZNF524 (ZNF524 BioID, ZNF524 KO proteome), and there is some discussion of the fact that ZNF524 interacts strongly with a protein involved in chromatin remodeling (DPY30), however these leads are not further explored.

Based on the reviewer's suggestion, we have now extended our BioID results. Our initial comparison of BirA*-ZNF524 WT vs. BirA*-ZNF524 ZF2 mut validated the *in vivo* association of ZNF524 with telomeres by yielding known telomeric proteins in proximity to ZNF524 WT. However, this comparison might have masked proteins that physically interact with ZNF524 independent of its ability to bind to telomeres. Therefore, we repeated the BioID experiment and this time compared BirA*-ZNF524 WT with BirA*-NLS (nuclear localization signal). Indeed, while we again recapitulate enrichment of TRF1, TRF2, NR2C1 and NR2C2 as telomeric proteins, we here identified additional candidate protein partners (**new Suppl. Fig. 3a-b, new Suppl. Table 5**). Potential ZNF524 interaction partners include ZMYM2 and ZMYM3, zinc finger proteins with recently described functions in homologous recombination (Lee et al., 2022, PMID: 35253893), and the three members of the chromatin remodelling ChAHP complex ADNP, CHD4 and CBX3/HP1gamma (Ostapcuk et al., 2018, PMID: 29795351).

We next focused on the ChAHP complex and screened for ChIP-grade antibodies for its complex members that would allow for enrichment of telomeric DNA and succeeded for CBX3. When comparing enrichment of telomeric DNA between U2OS WT and ZNF524 KO clones we quantified a ~2-fold lower enrichment in the ZNF524 KO clones (**new Suppl. Fig. 3c**). While these early data are not a definitive proof, they are in line with the notion that ZNF524-dependent epigenetic changes in telomeric chromatin might specifically impact the TRF2/RAP1 subcomplex.

Obviously, these additional findings are only the beginning of follow-up work that will require to carefully evaluate direct/physical interactions with the ChAHP complex or ZMYM2/3, and their detailed mechanistic influence on how ZNF524 impacts telomeres. We hope that the reviewer agrees with us that these experiments go beyond the scope of our current manuscript. Of note, we maintain that the identification of a protein factor specifically regulating the telomeric abundance of the TRF2/RAP1 subcomplex is an exciting finding to begin with. To our knowledge these data provide the first evidence in living cells of independent regulation and function of the TRF2/RAP1 subcomplex compared to the full 6-protein shelterin complex.

Minor points:

- Include a western blot confirming esiRNA knockdown.

Thank you for highlighting this oversight. We have now included ZNF524 esiRNA knock-down validation data (**new Suppl. Fig. 6e**).

- Line 220 refers to ZNF524 depletion, but the related figures are of knockout.

We have changed this to read “upon ZNF524 knock-out”.

Reviewer #3 (Remarks to the Author):

The manuscript by Braun et al described a zinc finger protein ZNF524 that locates on telomeric DNA to maintain telomere integrity. The authors have performed biochemical and structural analyses to demonstrate that ZNF524 directly binds telomeric repeats with high affinity *in vitro* and *in vivo* through its four zinc fingers for base-specific recognition. They also provide genetic evidence showing that ZNF524 maintains the TRF2/RAP1 subcomplex at telomeres thus preventing genomic instability. The biochemical and structural data are informative. However, some of the results obtained in the functional studies are not convincing, and may require more validations.

1. The authors claimed that ZNF524 directly binds to telomeric TTAGGG repeats with nanomolar affinity *in vitro*. Surprisingly, ZNF524 localizes to telomeres only in ALT cells but not in telomerase cells such as HeLa 1.3 that harbors long telomeres comparable to that in U2OS. What is the underlying mechanism that prevents ZNF524 from binding to telomeres in telomerase positive cells.

At present, we do not have an answer for this question. First, it remains possible that ZNF524 associates with telomeres in HeLa cells, but that with conventional detection methods (IF, ChIP) the signal/enrichment is not sufficient to exceed the background signal. In other words, 1-2 molecules of a transcription factor bound to a specific promoter are also not sufficient to yield discrete IF foci, but in return telomeres are much more abundant than single binding sites and hence generate more background signal in ChIP.

Regardless of these technical considerations, several of the proteins described in recent years to directly bind to telomeric DNA beyond the shelterin members have cell context-dependent degrees of telomere-association. For instance, similar to ZNF524, ZBTB48 (also known as TZAP) associates with half of all TRF2 foci in U2OS cells while association with telomeres in HeLa cells is only detectable at low frequency (Jahn et al., 2017, PMID: 28500257). Interestingly, Moreno et al. (PMID: 37013192) reported earlier this year that depletion of ATRX or DAXX in HeLa cells can boost telomeric association of overexpressed ZBTB48 from ~10% to >60% in ATRX- or DAXX-depleted cells. This illustrates that beyond simple competition with other direct telomere-binding proteins the epigenetic landscape at telomeres might impact the *in vivo* binding behaviour of these proteins and we agree that this is an important future direction.

2. Fig 3a and b. The authors pointed out that in >90% of U2OS cells co-localization of ZNF524 and TRF2 is visible with an average of 66% of all TRF2 foci being positive for ZNF524. A short statement of what TRF2 foci lack ZNF524 staining is desirable. Do they represent shorter telomeres? If not, can the authors think of anything that might have caused it?

We do not know which TRF2 foci specifically lack ZNF524 signal. As eluded in the previous points other protein factors such as ZBTB48, HMBOX1 (also known as HOT1) and NR2C2 show incomplete association with telomeres. The above mentioned manuscript on ATRX/DAXX depletion is the first explanation for any of these proteins (published during the course of this revision) and the mechanistic details of how exactly ATRX/DAXX depletion increase ZBTB48-telomere association are not yet clear.

There are obviously many other possible explanations (cell cycle specificity, dynamic association with different telomeres for their processing and/or dependence on individual chromatin state, competition with shelterin proteins and other direct telomere binding proteins, chromatin accessibility etc.) but we hope that the reviewer agrees that investigating this in mechanistic depth is a full project by itself.

3. Fig 3c and d. Given that ZNF524-GFP directly associates with more than half telomeres in a cell with high affinity, the ~2.5x enrichment of telomere DNA signal over background revealed by the ChIP assay implies that the IP efficiency is poor. Could it be due to the overexpression of ZNF524-GFP or the use of too few antibodies in the experiment? Would it be possible to perform the experiment using antibodies against endogenous ZNF524? In addition, ChIP with anti-TRF1/TRF2 is desired to serve as a positive control to better evaluate the enrichment of ZNF524.

While we would have preferred to perform ChIP with antibodies against endogenous ZNF524 (similar to our previous work on ZBTB48; Fig. 2 in Jahn et al., 2017, PMID: 28500257) the antibody that we raised for this project is not IP capable and only works in Western.

As eluded above, different assays come with different caveats of threshold intensities and background signal. We have here demonstrated *in vivo* association of ZNF524 with telomeres by IF, ChIP and BioID. We believe that the independent validation by three separate assays provides convincing evidence that ZNF524 binds to telomeres *in vivo*.

4. Fig 4a and b. The immunofluorescence staining of TRF1 in U2OS cells is of poor quality and seems not suitable for further quantitative analysis.

Without further details it is difficult for us to follow this comment. While the absolute signal intensity of TRF1 in the representative images is weaker compared to TRF2, overlapping telomeric foci are detected with little residual background in the rest of the nucleus. Therefore, it is unclear to us why exactly this reviewer deems this data of poor quality or insufficient for quantitative analysis. If at all, we would have been rather concerned with overexposure and signal saturation.

However, to further address this point, we have generated an independent dataset in which we have quantified the foci intensity of TRF1, TRF2 and RAP1 in the five independent U2OS WT and ZNF524 KO clones. As in the original manuscript, we again see that there is a significant reduction in TRF2 and RAP1 signal intensity at telomeres while TRF1 abundance is unaffected across the different clones. Fig. 4 contains more detailed information including matched complementation with the expression of either ZNF524 WT or ZF mutant constructs in all give ZNF524 KO clones. If the reviewers and editor do not deem this redundant, we can also offer to display this independent dataset as a supplementary figure.

Figure to the reviewers 1: (a) Representative immunofluorescence images of U2OS WT and ZNF524 KO cells stained for TRF2 (green) and TRF1 (red). Scale bars represent 10 μ m. Nuclei were counterstained with DAPI (blue). (b) Quantification of TRF1 and TRF2 IF signals in WT and KO clones. The violin plot shows the individual data points as densities. The mean is indicated by a solid line; *p*-values are determined by Student's *t*-test and indicated above the plot; ** *p*<0.01; *n*=5 for all conditions. (c) Representative IF pictures of U2OS WT and ZNF524 KO cells stained for RAP1 (red). (d) Quantification of the RAP1 IF signal in WT and KO clones depicted as violin plots. The mean is indicated by a solid line; *p*-values are determined by Student's *t*-test (*n*=5) and indicated above the plot; * *p*<0.05.

5. Fig 4b. The change in the density of TRF2 signal between WT group and KO group seems mild when compared to the intra-group variance caused by the high clonal heterogeneity. This makes it less convincing that depletion of ZNF524 leads to reduction in TRF2/subcomplex at telomere. Moreover, it seems that ZNF524 removal also causes a slight decrease in telomere length (Fig. S4f), which may be a reason for the mild change of TRF2 signal in the KO group.

While we see the reviewer's point, we respectfully disagree. The average changes of TRF2/RAP1 foci intensity across interphase and metaphase cells are up to 2-fold (please note the \log_2 scale for all IF quantifications). This would correspond to loss-of-heterozygosity for TRF2. As discussed in the manuscript these findings are in line with previously reported partial deprotection of telomeres/partial knock-down of TRF2 (Cesare et al., 2013, PMID: 23850488) and a quantitatively similar result was demonstrated upon Lamin B1 overexpression (Pennarun et al., 2021, PMID: 24469544). Therefore, our data demonstrates consequential changes in telomere biology in agreement with similar observations of multiple other research groups.

Furthermore, the observed changes in telomere length in U2OS ZNF524 KO clones (which are not significantly different from U2OS WT clones) are in the ballpark of 1-2kb (Supplementary Fig. 5e). Relative to an average telomere length of >20kb this does not explain the reduction in TRF2/RAP1 association with telomeres. Furthermore, we could not detect changes in POT1 and TRF1 foci intensity (Fig. 4a/b & Supplementary Fig. 6a/b). Therefore, we do not think that the reduction is simply the result of decreased telomere length.

6. Fig. S4e-h. Because of the high clonal heterogeneity, it seems inappropriate to directly compare the telomere length between WT and KO. Instead, comparison of telomere length among KO, KO+WT and KO+ZF2mut might be more informative.

We are somewhat surprised by this comment. Here, we have made the substantial effort to establish multiple individual WT and ZNF524 KO clones to rigorously compare genuine biological replicates. High clonal heterogeneity is a general concern with any CRISPR KO and needs to be systematically addressed as done in our study. While overexpression rescue experiments can be an important validation of loss-of-function experiments – as we have done for the quantification of TRF2 IF signals by reintroducing ZNF524 WT and ZNF524 ZF2 mut individually into each of our 5 independent ZNF524 KO clones (Fig. 4a-b) –, they are suboptimal if used alone and/or to “overwrite” statistically insignificant loss-of-function results. Given the existing heterogeneity in telomere length in cell culture populations, antibiotic selection for the different constructs could similarly introduce a selection bias especially when limited to single clones.

In this context, we would like to refer to a similar analysis in our previous work on ZBTB48 (Fig. 3A/B in Jahn et al., 2017, PMID: 28500257). Here despite the clonal heterogeneity a ZBTB48-dependent effect on telomere length could be established. This approach is in line with other publications in the field, e.g. Dille et al., 2016, PMID: 27760120.

7. The authors indicated that the function of ZNF524 is not restricted to ALT telomeres based on the observation that depletion of ZNF524 causes an increase in TIFs in both ALT (U2OS) and telomerase-positive (HeLa) cells. This observation is confusing since ZNF524 is not found at the telomere in HeLa cells. Will ZNF524 removal leads to decreased telomere-bound TRF2 and Rap1 in HeLa cells? What are the actual functions of ZNF524 in cells? Are the functions dependent on the telomere binding?

Please see above for considerations on protein abundance and detection thresholds. While we here describe one actual function of ZNF524, it is certainly possible that ZNF524 has additional roles (both telomeric and non-telomeric). While our manuscript focuses primarily on the function of ZNF524 at ALT telomeres, we agree that it will be important to further clarify any putative impacts in non-ALT cells in future work including for instance the differential abundance at telomeres as discussed in point 2 above.

Please note that based on the suggestion by reviewer 2, we have here further extended our previous BioID data. Our initial comparison of BirA*-ZNF524 WT vs. BirA*-ZNF524 ZF2 mut validated the *in vivo* association of ZNF524 with telomeres by yielding known telomeric proteins in proximity to ZNF524 WT. However, this comparison might have masked proteins that physically interact with ZNF524 independent of its ability to bind to telomeres. Therefore, we repeated the BioID experiment and this time compared BirA*-ZNF524 WT with BirA*-NLS (nuclear localization signal). Indeed, while we again recapitulate enrichment of TRF1, TRF2, NR2C1 and NR2C2 as telomeric proteins, we here identified additional candidate protein partners (**new Suppl. Fig. 3a-b, new Suppl. Table 5**). Potential ZNF524 interaction partners include ZMYM2 and ZMYM3, zinc finger proteins with recently described functions in homologous recombination (Lee et al., 2022, PMID: 35253893), and the three members of the chromatin remodelling ChAHP complex ADNP, CHD4 and CBX3/HP1gamma (Ostapcuk et al., 2018, PMID: 29795351).

We next focused on the ChAHP complex and screened for ChIP-grade antibodies for its complex members that would allow for enrichment of telomeric DNA and succeeded for CBX3. When comparing enrichment of telomeric DNA between U2OS WT and ZNF524 KO clones we quantified a ~2-fold lower enrichment in the ZNF524 KO clones (**new Suppl. Fig. 3c**). While these early data are not a definitive proof, they are in line with the notion that ZNF524-dependent epigenetic changes in telomeric chromatin might specifically impact the TRF2/RAP1 subcomplex.

Obviously, these additional findings are only the beginning of follow-up work that will require to carefully evaluate direct/physical interactions with the ChAHP complex or ZMYM2/3, and their detailed mechanistic impact on how ZNF524 impacts telomeres. We hope that the reviewer agrees with us that these experiments go beyond the scope of our current manuscript. Of note, we maintain that the identification of a protein factor specifically regulating the telomeric abundance of the TRF2/RAP1 subcomplex is an exciting finding to begin with. To our knowledge these data provide the first evidence in living cells of independent regulation and function of the TRF2/RAP1 subcomplex compared to the full 6-protein shelterin complex.

REVIEWER COMMENTS

Reviewer #2 (Remarks to the Author):

The authors have done a good job in addressing the reviewers comments. In particular the addition of further BioID experiments adds to the manuscript. I appreciate the comments regarding TRF2/RAP1 levels and telomere length, and also the effort put in to generating an additional set of data for the quantitation of TRF2/RAP1/TRF1 foci; however, I do feel that (as in my original review) TRF2/RAP1 quantitation relative to telomere probe with/without ZNF524 should be included to more directly demonstrate the reduction in TRF2/RAP1 binding to telomeres in the absence of ZNF524.

Reviewer #3 (Remarks to the Author):

The authors have addressed most of my comments, and I have only one comment left. In this paper, a major conclusion is that ZNF524 mediates TRF2 and RAP1 localization to telomeres (Fig 4a and b). Despite the high clonal heterogeneity shown within both the WT group and KO group, the authors claimed that there is a significant decrease in TRF2 signal intensity in the KO group compared to the WT group (2-fold, * $p < 0.05$). However, among the five KO clones, introducing the ZNF524 ZF2mut into two of them (KO1 and KO2) likely rescues the reduction of TRF2 focal intensity, similar to that of ZNF524 WT (judged by the mean values). This may not be simply attributed to clone heterogeneity. If the authors believe that it is caused by differences in protein expression or antibiotic selection, incorporating replicated results would be helpful.

REVIEWER COMMENTS (BLACK)

AUTHOR RESPONSE (BLUE)

We would again like to thank all reviewers for their additional constructive comments and their overall positive feedback on our revised manuscript.

Reviewer #2 (Remarks to the Author)

The authors have done a good job in addressing the reviewers comments. In particular the addition of further BioID experiments adds to the manuscript. I appreciate the comments regarding TRF2/RAP1 levels and telomere length, and also the effort put in to generating an additional set of data for the quantitation of TRF2/RAP1/TRF1 foci; however, I do feel that (as in my original review) TRF2/RAP1 quantitation relative to telomere probe with/without ZNF524 should be included to more directly demonstrate the reduction in TRF2/RAP1 binding to telomeres in the absence of ZNF524.

We understand the reasoning of the reviewer. However, the manuscript already includes a separate qFISH quantification, which shows a (non-significant) <10% reduction in telomeric FISH intensity in U2OS WT compared to ZNF524 KO clones. Importantly, immunoFISH has its own technical caveats, e.g. the rather harsh treatments needed for DNA denaturation can impact the IF staining. As the reviewer acknowledged, we already quantified the reduction in TRF2 and RAP1 IF foci comparing 5 U2OS WT and ZNF524 KO clones (= 5 biological replicates) in two completely independent datasets (Fig. 4 & Figure to the reviewers 1 from the first round of reviews). Here, TRF2/RAP1 foci intensity had been robustly reduced across the ZNF524 KO clones while TRF1 foci intensity remained unchanged. The latter is again supporting our qFISH data. Therefore, repeating both the IF and FISH stainings yet another time while introducing new technical caveats by combining both procedures will not change the main conclusion of our manuscript.

Reviewer #3 (Remarks to the Author)

The authors have addressed most of my comments, and I have only one comment left. In this paper, a major conclusion is that ZNF524 mediates TRF2 and RAP1 localization to telomeres (Fig 4a and b). Despite the high clonal heterogeneity shown within both the WT group and KO group, the authors claimed that there is a significant decrease in TRF2 signal intensity in the KO group compared to the WT group (2-fold, * p<0.05). However, among the five KO clones, introducing the ZNF524 ZF2 mut into two of them (KO1 and KO2) likely rescues the reduction of TRF2 focal intensity, similar to that of ZNF524 WT (judged by the mean values). This may not be simply attributed to clone heterogeneity. If the authors believe that it is caused by differences in protein expression or antibiotic selection, incorporating replicated results would be helpful.

Our dataset is based on five independent knock-out clones as rigorous biological replicates. The suggestion of re-transducing each KO clone at least 3 times (which would inflate the dataset in Fig. 4b to 30 parallel cultures) to control for variation in the antibiotic selection and to test whether each clone individually shows the expected rescue with ZNF524 WT and lack of rescue with ZNF524 ZF2 mut seems excessive in comparison to what we think was already an above average effort in rigor. Importantly, based on the statistics of 5 biological replicates / 5 independent knock-out clones, the ZNF524 WT construct shows a significant gain in TRF2 foci intensity compared to both non-transduced knock-out clones and the same knock-out clones expressing the ZNF524 ZF2 mut construct.

In addition, we would like to reiterate that we had also observed reduced TRF2 abundance on native metaphase spreads upon transient ZNF524 knock-down (**Suppl. Fig. 6c-e**). We are now highlighting more clearly that this is corroborating evidence based on “cells that had not undergone clonal expansion” (**page 7, line 201**).